# FaCE: Provable Frequency-Aware Convex Enhancement for Training-Free Low-Light Images

## Abstract

We study training-free low-light image enhancement from a convex optimization viewpoint with theoretical guarantees. Building on the Monogenic Fourier Transform (MFT), we introduce Frequency-Aware Convex Enhancement (FaCE), a frequency-aware convex enhancement method for training-free low-light images with guarantees on existence, uniqueness, and stability of the per-image solution. To make the frequency modeling reproducible, we (i) define the low-pass prior via the spectral centroid $(u_c, v_c)$ and an energy–cumulative distribution radius $r_\tau$ (the smallest radius achieving cumulative spectral energy $\tau$), and (ii) select the number of spectral clusters $K$ with a data-driven model-selection rule. This yields a fully training-free pipeline with clear variables and no learned parameters. We prove that a unique, stable solution exists and verify the guarantees via numerical experiments. On standard benchmarks, FaCE attains competitive quality with a per-image solver, and we include in-the-wild qualitative mosaics on real images to highlight practical usefulness. Rather than competing with large learned priors, FaCE complements them as a theoretically grounded, interpretable alternative that requires no training and exposes frequency-band attributions.

## 1 Introduction

Low-light image enhancement (LLIE) is often handled by heuristics or supervised models that depend on training and ad-hoc choices, with no guarantees of existence, uniqueness, or stability. We instead cast LLIE as a Monogenic Fourier Transform (MFT)-grounded (Felsberg & Sommer, 2002; Unser et al., 2009), training-free, per-image strictly convex optimization and prove well-posedness; the solver uses no learned parameters.

We make frequency modeling reproducible: the low-pass prior is set by the spectral centroid $(u_c, v_c)$ and energy-CDF (Cumulative Distribution Function) radius $r_\tau$; a spectral gain map $W$ updates illumination only, and inverse MFT reconstructs the output. The result is a theoretically grounded, interpretable alternative with clear in-the-wild visuals. As illustrated in Figure 1, FaCE turns frequency-band energy maps into convex brightness and structure gains in the image domain, yielding an exposure-controlled enhanced result.

Our contributions are threefold:

- Grounded in MFT, we cast LLIE as a strictly convex per-image optimization and prove well-posedness (existence, uniqueness, stability) with a solver that uses no learned parameters.
- A low-pass prior defined by $(u_c, v_c)$ and the energy-CDF radius $r_\tau$, together with data-driven diagnostics for $K$, replaces subjective spectrum cuts with principled, transparent guidance.
- We derive band-level contribution maps directly from the convex objective, exposing the frequency-to-image pathway, and reveal a principled dependence on $K$ and $r_\tau$.

The paper is organized as follows: Section 2 reviews related work, Section 3 introduces FaCE, Section 4 presents experiments, and Section 5 concludes.

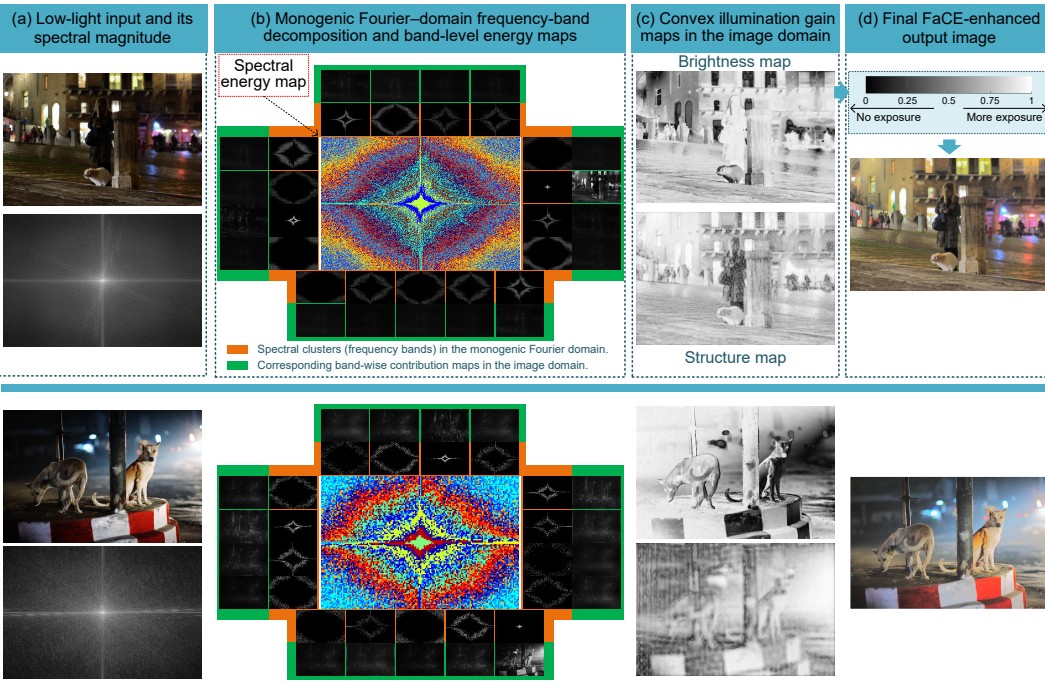

Figure 1: Visual overview of the FaCE frequency-aware convex enhancement pipeline. (a) A low-light input image (from ExDark dataset) and its spectral magnitude. (b) Monogenic Fourier-domain band decomposition and the corresponding band-level energy maps. (c) Convex illumination gain maps in the image domain, including the brightness map and structure map. (d) Final FaCE-enhanced output image.

## 2    RELATED WORKS

### 2.1    LIMITATIONS OF SUPERVISED DEEP-LEARNING METHODS

In recent years, supervised deep-learning approaches, such as Convolutional Neural Networks (CNNs) and transformer-based methods (e.g., UHD Transformer, (Wang et al., 2024a)), have achieved notable success in low-light enhancement tasks. These methods treat the enhancement problem as a regression task using paired low-light and normal-light images, heavily relying on large-scale annotated datasets. Despite their empirical effectiveness, supervised methods lack rigorous mathematical guarantees concerning critical solution properties such as stability and uniqueness, resulting in poor generalization beyond training datasets (Wang et al., 2023a). Moreover, this dependency on extensive labeled data significantly restricts their interpretability and practical applicability in real-world scenarios.

### 2.2    EMERGING UNSUPERVISED AND FREQUENCY-DOMAIN APPROACHES

Recently, unsupervised approaches have emerged to address the dependency on labeled data, such as Zero-DCE (Guo et al., 2020), EnlightenGAN (Jiang et al., 2021), and ExposureDiffusion (Wang et al., 2023b). These methods estimate illumination curves without supervision, yet they commonly lack rigorous mathematical formulations, particularly neglecting explicit frequency-domain analyses. More recent frequency-domain methods—including UHDFour (Li et al.), FourierDiff (Lv et al., 2024), ULEFD (Ming et al., 2023), and DPLUT (Lin et al., 2025)—use Fourier-domain techniques, diffusion priors, or lookup tables. Although these approaches demonstrate improved visual performance, they typically fall short in providing rigorous mathematical proofs regarding the existence, uniqueness, and stability of their solutions, which are essential for robust interpretation and reliability.

## 2.3 RESEARCH MOTIVATION AND PROBLEM FOCUS

Classical Retinex- and filtering-based pipelines (Land & McCann, 1971; Jobson et al., 1997; Fu et al., 2016) remain attractive for their simplicity and interpretability, but typically rely on heuristic choices and lack end-to-end guarantees. Supervised deep models (CNN/Transformer) deliver strong perceptual quality at the cost of dataset-level training and limited theoretical guarantees. Recent training-free and frequency-domain approaches (e.g., Zero-DCE, EnlightenGAN; Fourier- and diffusion/LUT-based methods) reduce annotation demands yet still provide few statements about well-posedness. Our work is complementary: a training-free, frequency-aware formulation with provable convexity and reproducible spectral modeling, positioned as an interpretable alternative rather than a parameter-hungry competitor.

## 3 METHODOLOGY

As shown in Figure 2, we cast LLIE as an MFT-grounded frequency-domain inverse optimization—proving existence, uniqueness, and stability; deriving a discrete computational form; and constructing an illumination-only spectral gain map via spectral clustering.

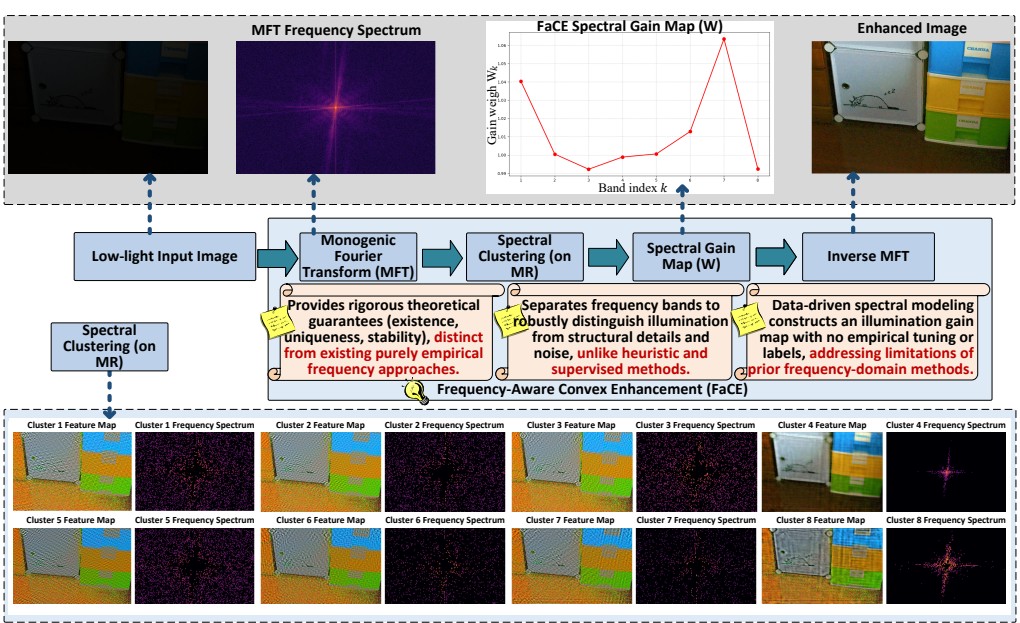

Figure 2: FaCE builds a spectral gain map via MFT clustering and enhances illumination without learned parameters.

## 3.1 INVERSE OPTIMIZATION PROBLEM FOR LLIE

Low-light image enhancement is fundamentally an inverse optimization problem of recovering high-quality images from degraded low-illumination observations. Formally, we consider an observed image defined as a square-integrable function in a two-dimensional spatial domain $\Omega \subseteq \mathbb{R}^2$:

$$I \colon \Omega \to [0,1], \ I \in L^2(\Omega). \tag{1}$$

According to the classical Retinex theory proposed by Land & McCann (1971), any observed image $I(x, y)$ can be strictly represented as the product of a reflectance component $R(x, y)$ and an illumination component $L(x, y)$:

$$I(x,y) = R(x,y) \cdot L(x,y), \quad R(x,y), L(x,y) \in [0,1], \tag{2}$$

here $R(x, y)$ denotes reflectance intrinsic to object surfaces, and $L(x, y)$ represents spatial illumination distribution.

We assume illumination $L(x, y)$ is smooth and slowly varying ($L(x, y) \in C^\infty(\Omega) \cap [0, 1]$), while reflectance $R(x, y)$ contains edges and high-frequency details.

The primary objective of low-light image enhancement is to accurately enhance the illumination component $L(x, y)$. Mathematically, given an observed low-light image $I(x, y)$, the enhancement problem can be formalized as seeking an operator $\mathcal{T}$ designed to enhance the illumination:

$$L_{enh}(x, y) = \mathcal{T}[L(x, y)], \tag{3}$$

such that

$$L_{enh}(x, y) > L(x, y), \quad \forall (x, y) \in \Omega. \tag{4}$$

Throughout this study, we treat reflectance as invariant and optimize illumination only: $R_{enh}(x, y) \equiv R(x, y)$, and $I_{enh}(x, y) = R(x, y)L_{enh}(x, y)$.

From an inverse optimization problem perspective, this task involves reconstructing a stable and clearly defined enhanced illumination component $L_{enh}(x, y)$ from the observed image $I(x, y)$. However, this inverse optimization problem is intrinsically ill-posed, characterized by fundamental mathematical challenges: (1) Existence: An illumination component satisfying the enhancement conditions may not always exist for every observed image. (2) Uniqueness: Multiple valid illumination solutions might exist under typical conditions. (3) Stability: Small perturbations in observed data can significantly alter enhancement outcomes.

To address the absence of rigorous guarantees in Retinex heuristics and supervised methods, we introduce FaCE—a training-free, per-image convex formulation in the frequency domain that proves well-posedness and uses reproducible spectral modeling.

## 3.2 MFT and Spectral Definition

Unlike conventional Fourier features, MFT provides rotation-covariance, directionally sensitive spectra that cleanly characterize illumination and structure, making it well suited for low-light analysis and for the rigorous formulation we develop.

To analyze the frequency-domain characteristics inherent in low-light enhancement, we introduce the two-dimensional MFT. We recall the two-dimensional MFT:

For any function $f(x, y)$, the two-dimensional MFT is defined as:

$$\mathcal{M}\{f(x, y)\}(u, v) = \left( \hat{f}(u, v), \; -i \, \frac{u}{\sqrt{u^2 + v^2}} \, \hat{f}(u, v), \; -i \, \frac{v}{\sqrt{u^2 + v^2}} \, \hat{f}(u, v) \right), \tag{5}$$

with $(u, v) \neq (0, 0)$ and the condition $\mathcal{M}\{f\}(0, 0) = \left( \hat{f}(0, 0), 0, 0 \right)$, ensuring mathematical consistency and rigorous definition, and $\hat{f} = \mathcal{F}\{f\}$ is the two-dimensional Fourier transform. The more details about the MFT, refer to Felsberg & Sommer (2002); Unser et al. (2009).

## 3.3 Frequency-Aware Convex Enhancement

To address the inverse optimization problem defined in Section 3.1, we propose a mathematically grounded approach termed FaCE. FaCE uses the spectral-domain representation derived from the MFT, providing a mathematically interpretable framework to separate illumination from reflectance in the frequency domain.

We begin by considering the MFT spectral representation of an observed low-light image $I(x, y)$. Define the MFT magnitude spectrum as:

$$M_R(u, v) = \log(1 + |\mathcal{M}\{I(x, y)\}(u, v)|), \quad (u, v) \in \Omega_f \subseteq \mathbb{R}^2. \tag{6}$$

Here $|\cdot|$ denotes the Euclidean norm across the three MFT channels.

Given that illumination predominantly resides in low-frequency components, and reflectance structures and noise occupy distinct frequency bands, we construct a frequency-domain clustering to rigorously capture these intrinsic properties. Formally, we define a frequency-domain partitioning problem as follows:

$$\{C_k\}_{k=1}^K = \arg \min_{\{C_k\}} \sum_{k=1}^K \iint_{C_k} |M_R(u, v) - m_k|^2 \, du \, dv, \quad k \in \mathbb{N}^*, \tag{7}$$

subject to the constraints $C_i \cap C_j = \emptyset$, $\bigcup_{k=1}^{K} C_k = \Omega_f$, where the local cluster mean $m_k$ and the global spectral mean $m_{glo}$ are rigorously defined as:

$$m_k = \frac{\iint_{C_k} M_R(u,v)\, du\, dv}{|C_k|}, \quad m_{glo} = \frac{\iint_{\Omega_f} M_R(u,v)\, du\, dv}{|\Omega_f|}. \tag{8}$$

The construction of these frequency-domain clusters provides a mathematical interpretation of illumination and structure separation in the frequency domain, directly addressing interpretability issues inherent in traditional heuristic and supervised methods.

We then define a frequency-aware spectral gain map $W(u,v)$ from the derived clusters:

$$W(u,v) = \sum_{k=1}^{K} [\alpha + \beta(m_k - m_{glo})] \cdot \mathbb{I}_{C_k}(u,v), \tag{9}$$

with the indicator function rigorously defined as:

$$\mathbb{I}_{C_k}(u,v) = \begin{cases} 1, & (u,v) \in C_k, \\ 0, & otherwise. \end{cases} \tag{10}$$

Parameters $\alpha, \beta \in \mathbb{R}$ are determined within a training-free optimization framework, thereby ensuring mathematically rigorous and interpretable illumination gain map.

To further stabilize the solution and limit the influence of high-frequency noise, we incorporate a mathematically defined low-pass filter:

$$LP(u,v) = \begin{cases} 1, & (u-u_c)^2 + (v-v_c)^2 \leq r_\tau{}^2, \\ 0, & otherwise, \end{cases} \tag{11}$$

where the parameters $(u_c, v_c)$ and radius $r_\tau$ ($\tau$ is cumulative spectral energy) are chosen according to the frequency-domain illumination characteristics.

Thus, the enhanced illumination component $L_{enh}(x,y)$ is robustly reconstructed by the inverse MFT, integrating data-driven weighting and low-pass filtering to ensure stable, interpretable, and visually superior illumination enhancement:

$$\begin{aligned} L_{enh} &= \left\{ \mathcal{M}^{-1} \left( \mathcal{M} \cdot W \cdot LP \right) \right\} \cdot L \\ &:= \mathcal{T} \cdot L. \end{aligned} \tag{12}$$

Thus, identifying the enhancement operator in Eq. (3) as

$$\mathcal{T} = \left\{ \mathcal{M}^{-1} \left( \mathcal{M} \cdot W \cdot LP \right) \right\}. \tag{13}$$

We perform a luminance–chroma decomposition with $L = \text{luminance}(I)$ and $R = I/(L + \sigma)$; we enhance luminance only to get $L_{enh}$ and reconstruct $I_{enh} = R \cdot L_{enh}$ (see Appx. A for details).

### 3.4 Theoretical Analysis: Existence, Uniqueness, and Stability

In this subsection, we rigorously establish the mathematical properties—existence, uniqueness, and stability—of the solution to the FaCE optimization problem formulated in Section 3.3. To do this, we provide definitions, conditions, and a detailed mathematical proof.

Consider the optimization problem defined as follows:

$$\min_{\alpha,\beta} \mathcal{E}(I_{enh}, I) = \|I_{enh} - I\|_{L^2(\Omega)}^2 + \lambda_1 \mathcal{E}_{sc}(I_{enh}, I) + \lambda_2 \mathcal{E}_{exp}(I_{enh}), \quad \lambda_1, \lambda_2 > 0, \tag{14}$$

and the individual terms of the functional are defined as:

*Data fidelity term:*

$$\|I_{enh} - I\|_{L^2(\Omega)}^2 = \iint_\Omega |I_{enh}(x,y) - I(x,y)|^2 \, dx\, dy. \tag{15}$$

*Spatial consistency loss:*

$$\mathcal{E}_{sc}(I_{enh}, I) = \iint_\Omega \left| \nabla^2 I_{enh}(x,y) - \nabla^2 I(x,y) \right|^2 dx\, dy. \tag{16}$$

*Exposure control loss:*

$$\mathcal{E}_{exp}(I_{enh}) = \left| \frac{1}{|\Omega|} \iint_\Omega I_{enh}(x,y)\, dx\, dy - E_{target} \right|^2. \tag{17}$$

**Main Theorem.** *If $\mathcal{E}(I_{enh}, I)$ is strictly convex in the solution space $L^2(\Omega)$ and $\mathcal{E}(I_{enh}, I)$ is lower semi-continuous on the function space $L^2(\Omega)$, then the FaCE optimization problem admits a unique global minimizer $(\alpha^*, \beta^*)$, ensuring solution existence, uniqueness, and stability.*

*Proof.* We prove the theorem by following these rigorous steps:

(1) **Existence of the Solution.** Consider the functional:

$$\mathcal{E}(I_{enh}, I) = \|I_{enh} - I\|_{L^2(\Omega)}^2 + \lambda_1 \mathcal{E}_{sc}(I_{enh}, I) + \lambda_2 \mathcal{E}_{exp}(I_{enh}). \tag{18}$$

Since the domain $L^2(\Omega)$ is a Hilbert space, we first show that each term of the functional is convex and lower semi-continuous:

The data fidelity term $\|I_{enh} - I\|_{L^2(\Omega)}^2$ is quadratic and thus strictly convex and continuous in the Hilbert space $L^2(\Omega)$.

The spatial consistency loss $\mathcal{E}_{sc}(I_{enh}, I)$, defined using the second-order differential operator $\nabla^2$, corresponds to a norm-based difference, which is convex and continuous in the appropriate Sobolev space (embedding into $L^2(\Omega)$).

The exposure control loss $\mathcal{E}_{exp}(I_{enh})$ involves averaging operations and quadratic terms, thus also explicitly convex and continuous.

Given that a sum of strictly convex and lower semi-continuous functions with strictly positive weights $\lambda_1, \lambda_2$ remains strictly convex and lower semi-continuous, the functional $\mathcal{E}(I_{enh}, I)$ satisfies these conditions.

By Ekeland & Temam (1999); Dacorogna (2008), $\mathcal{E}(I_{enh}, I)$ admits a minimizer $I_{\text{enh}}^* \in L^2(\Omega)$. Writing $I_{enh}(\alpha, \beta) = \alpha U_\alpha + \beta U_\beta$, the parameter-space Hessian $H_{ij} = \langle U_i, U_j \rangle$ is symmetric positive definite when $U_\alpha, U_\beta$ are linearly independent; hence $\mathcal{E}(\alpha, \beta)$ is strongly convex with a unique minimizer $(\alpha^*, \beta^*)$ and Lipschitz stability (details in Appx. B).

(2) **Uniqueness of the Solution.** To establish uniqueness, we consider two hypothetical solutions $I_{enh}^{(1)}$ and $I_{enh}^{(2)}$ both minimizing $\mathcal{E}$. Assume by contradiction $I_{enh}^{(1)} \neq I_{enh}^{(2)}$. Define a convex combination:

$$I_{enh}^{(\gamma)} = \gamma I_{enh}^{(1)} + (1-\gamma) I_{enh}^{(2)}, \quad 0 < \gamma < 1. \tag{19}$$

Given strict convexity of $\mathcal{E}(I_{enh}, I)$, we have:

$$\mathcal{E}(I_{enh}^{(\gamma)}, I) < \gamma \mathcal{E}(I_{enh}^{(1)}, I) + (1-\gamma)\mathcal{E}(I_{enh}^{(2)}, I). \tag{20}$$

But since $I_{enh}^{(1)}$ and $I_{enh}^{(2)}$ are both minimizers, the above inequality contradicts their optimality. Thus, the solution must be unique (details in Appx. C).

(3) **Stability of the Solution.** Stability is ensured by parameter-space strong convexity with modulus $\mu = \lambda_{\min}(H)$; the solution map is Lipschitz with constant $O(1/\mu)$. Given the functional's quadratic structure and strict convexity assumptions, its Hessian is strictly positive definite. Consequently, minor perturbations in the observed image $I(x,y)$ result in bounded and stable solution variations, confirming solution stability (details in Appx. D).

Under strict convexity and lower semicontinuity, FaCE admits a unique global minimizer that is stable to small perturbations. $\qquad\square$

### 3.5 DISCRETE APPROXIMATION AND NUMERICAL IMPLEMENTATION

To apply the FaCE method practically, we discretize the continuous-domain formulation for numerical computation. Consider the discrete representation of an observed image $I[m,n]$, defined on a discretized spatial domain $\Omega_d = \{(m,n) \mid m = 0, \ldots, H-1;\ n = 0, \ldots, W-1\}$, where $d = H \times W$ is the image resolution.

#### 3.5.1 DISCRETE MFT

The discrete two-dimensional MFT is defined as:

$$\mathcal{M}\{I\}[u,v] = \left( \hat{I}[u,v],\ -i\frac{u}{\rho[u,v]}\hat{I}[u,v],\ -i\frac{v}{\rho[u,v]}\hat{I}[u,v] \right),\ \rho = \sqrt{u^2 + v^2}, \tag{21}$$

where $i$ is the imaginary unit, and the definition at the singularity $\mathcal{M}(0,0) = \left( \hat{I}(0,0),\ 0,\ 0 \right)$.

#### 3.5.2 DISCRETE FREQUENCY-AWARE DATA-DRIVEN SPECTRAL CLUSTERING

We define the discrete spectral magnitude of the MFT as:

$$M_R[u,v] = \log\left(1 + |\mathcal{M}\{I[m,n]\}[u,v]|\right). \tag{22}$$

We formulate the discrete frequency-domain spectral clustering optimization problem as follows:

$$\{C_k\}_{k=1}^K = \arg\min_{\{C_k\}} \sum_{k=1}^K \sum_{(u,v) \in C_k} \left| M_R[u,v] - m_k \right|^2, \tag{23}$$

with constraints $C_i \cap C_j = \emptyset$, $\bigcup_{k=1}^K C_k = \Omega_f^d$, where the discrete local and global spectral means are calculated as:

$$m_k = \frac{\sum_{(u,v) \in C_k} M_R[u,v]}{|C_k|},\quad m_{glo} = \frac{\sum_{(u,v) \in \Omega_f^d} M_R[u,v]}{|\Omega_f^d|}. \tag{24}$$

#### 3.5.3 DISCRETE FREQUENCY-AWARE CONVEX ENHANCEMENT

Given the discrete spectral clusters $\{C_k\}$, the frequency-aware gain map is:

$$W[u,v] = \sum_{k=1}^K \left[\alpha + \beta(m_k - m_{glo})\right] \cdot \mathbb{I}_{C_k}[u,v], \tag{25}$$

where the indicator function is:

$$\mathbb{I}_{C_k}[u,v] = \begin{cases} 1, & (u,v) \in C_k, \\ 0, & otherwise. \end{cases} \tag{26}$$

We further define the discrete low-pass filter as:

$$LP[u,v] = \begin{cases} 1, & (u - u_c)^2 + (v - v_c)^2 \le r_\tau^2, \\ 0, & otherwise, \end{cases} \tag{27}$$

where the center frequency coordinates $(u_c, v_c)$ and radius $r_\tau$ are chosen based on image frequency.

#### 3.5.4 NUMERICAL OPTIMIZATION AND IMPLEMENTATION

The enhanced image illumination component in discrete form is obtained through the inverse MFT:

$$L_{enh}[m,n] = \left\{ \mathcal{M}^{-1}(\mathcal{M} \cdot W \cdot LP) \right\} L[m,n]. \tag{28}$$

We optimize parameters $\alpha, \beta$ by numerically minimizing the discrete counterpart of the continuous-domain optimization functional:

$$\mathcal{E}(I_{enh}, I) = \|I_{enh} - I\|_F^2 + \lambda_1 \mathcal{E}_{sc}(I_{enh}, I) + \lambda_2 \mathcal{E}_{exp}(I_{enh}), \tag{29}$$

Table 1: PSNR and SSIM comparison on the LOL-v1 and LOL-v2-Real datasets between existing methods and the proposed FaCE.

| Method | Source | Setting | Params (M)↓ | LOL-v1 | | LOL-v2-Real | |
|---|---|---|---|---|---|---|---|
| | | | | PSNR↑ | SSIM↑ | PSNR↑ | SSIM↑ |
| EnlightenGAN | IEEE TIP 2021 | UL-U | 8.44 | 17.48 | 0.65 | 18.64 | 0.68 |
| CLIP-LIT | ICCV 2023 | UL-U | 0.28 | 17.21 | 0.59 | 17.06 | 0.59 |
| LightenDiffusion | ECCV 2024 | UL-U | 27.81 | **20.45** | 0.80 | **20.60** | **0.81** |
| Zero-DCE | CVPR 2020 | UL-ZR | 0.079 | 14.86 | 0.56 | 18.06 | 0.57 |
| RUAS | CVPR 2021 | UL-ZR | 0.01 | 16.40 | 0.50 | 15.33 | 0.49 |
| SCI | CVPR 2022 | UL-ZR | 0.0003 | 14.78 | 0.52 | 17.30 | 0.53 |
| DEnet | ICLR 2025 | UL-ZR | 0.35 | 19.80 | 0.75 | 20.22 | 0.79 |
| QuadPrior | CVPR 2024 | ZR-T | 1313.60 | 20.31 | 0.81 | 20.59 | **0.81** |
| ExCNet | ACMMM 2019 | ZS | 8.27 | 13.88 | 0.65 | 15.50 | 0.57 |
| GDP | CVPR 2023 | ZS | 552.81 | 15.83 | 0.69 | 14.36 | 0.63 |
| LLIEDiff | AAAI 2025 | ZS | 1066.24 | 19.82 | **0.84** | 19.95 | 0.78 |
| JieP | ICCV 2017 | 0P–0T | **0.00** | 11.58 | 0.48 | – | – |
| MSR | Neuroc 2017 | 0P–0T | **0.00** | 15.91 | 0.40 | 14.90 | 0.41 |
| **FaCE (Ours)** | – | 0P–0T | **0.00** | 17.04 | 0.73 | 17.17 | 0.70 |

–: Not reported; UL-U: Unpaired Learning; UL-ZR: Zero-Reference Trained; ZR-T: Zero-Reference—Trained on Normal-only; ZS: Zero-Shot Inference; 0P–0T: Parameter-free & Training-free.

Table 2: ExDark detection results with different low-light enhancement preprocessors.

| Method | Source | Setting | Params (M)↓ | ExDark (Bicycle, Car, Chair, Dog, People,...) | | | | |
|---|---|---|---|---|---|---|---|---|
| | | | | NIQE↓ | BRISQUE↓ | PIQE↓ | mAP↑ | $AP_{50}$↑ |
| DEnet | ICLR 2025 | UL-ZR | 3.45 | **3.914** | **27.577** | 38.582 | **0.633** | **0.892** |
| CLODE | ICLR 2025 | UL-ZR | 0.29 | 4.521 | 37.137 | 46.677 | 0.623 | 0.888 |
| DarkIR | CVPR 2025 | Sup-P | 3.32 | 4.060 | 29.631 | 41.554 | 0.626 | 0.891 |
| **FaCE (Ours)** | Ours | 0P–0T | **0.00** | 4.343 | 30.071 | **36.142** | 0.600 | 0.878 |

UL-ZR: Zero-Reference Trained; Sup-P: Supervised—Trained on paired low/normal (or degraded/clean) images; 0P–0T: Parameter-free & Training-free.

where the discrete forms of spatial consistency and exposure control losses are:

$$\mathcal{E}_{sc}(I_{enh}, I) = \sum_{m,n} \left| \nabla^2 I_{enh}[m,n] - \nabla^2 I[m,n] \right|^2, \tag{30}$$

$$\mathcal{E}_{exp}(I_{enh}) = \left| \frac{1}{HW} \sum_{m,n} I_{enh}[m,n] - E_{target} \right|^2. \tag{31}$$

To numerically solve this training-free optimization problem, we employ gradient-based iterative methods to minimize the defined energy functional (Eq. (29)). The optimization process is completely training-free, relying solely on intrinsic image information rather than external labeled data. By implementing these discrete methods, FaCE achieves mathematically rigorous, computationally stable, and visually robust low-light enhancement.

## 4 EXPERIMENTAL EVALUATION

We extensively evaluate FaCE on the real-world LOL-v1, LOL-v2-Real benchmarks (Wei et al., 2018), as well as the night-time datasets ExDark (Loh & Chan, 2019), and DarkFace (Wang et al., 2021), and the exposure dataset SICE (Cai et al., 2018). We compare against a wide range of LLIE baselines, including Retinex-based, zero-reference, GAN/Transformer-based, and diffusion-based methods (e.g., Zero-DCE, RUAS (Liu et al., 2021), SCI (Ma et al., 2022), DEnet (Huaqiu et al., 2025), EnlightenGAN, CLIP-LIT (Liang et al., 2023), LightenDiffusion (Jiang et al., 2024), DarkIR (Feijoo et al., 2025), CLODE (Jung et al., 2025), QuadPrior (Wang et al., 2024b), GDP (Fei et al., 2023), ExCNet (Zhang et al., 2019), LLIEDiff (Huang et al., 2025), JieP (Cai et al., 2017), MSR (Zhang et al., 2017)). We further run dedicated experiments to validate our theory: (1) numerical stability under perturbations; (2) the effectiveness of the frequency-aware spectral clustering; and (3) empirical confirmation of existence, uniqueness, and stability of the convex solution.

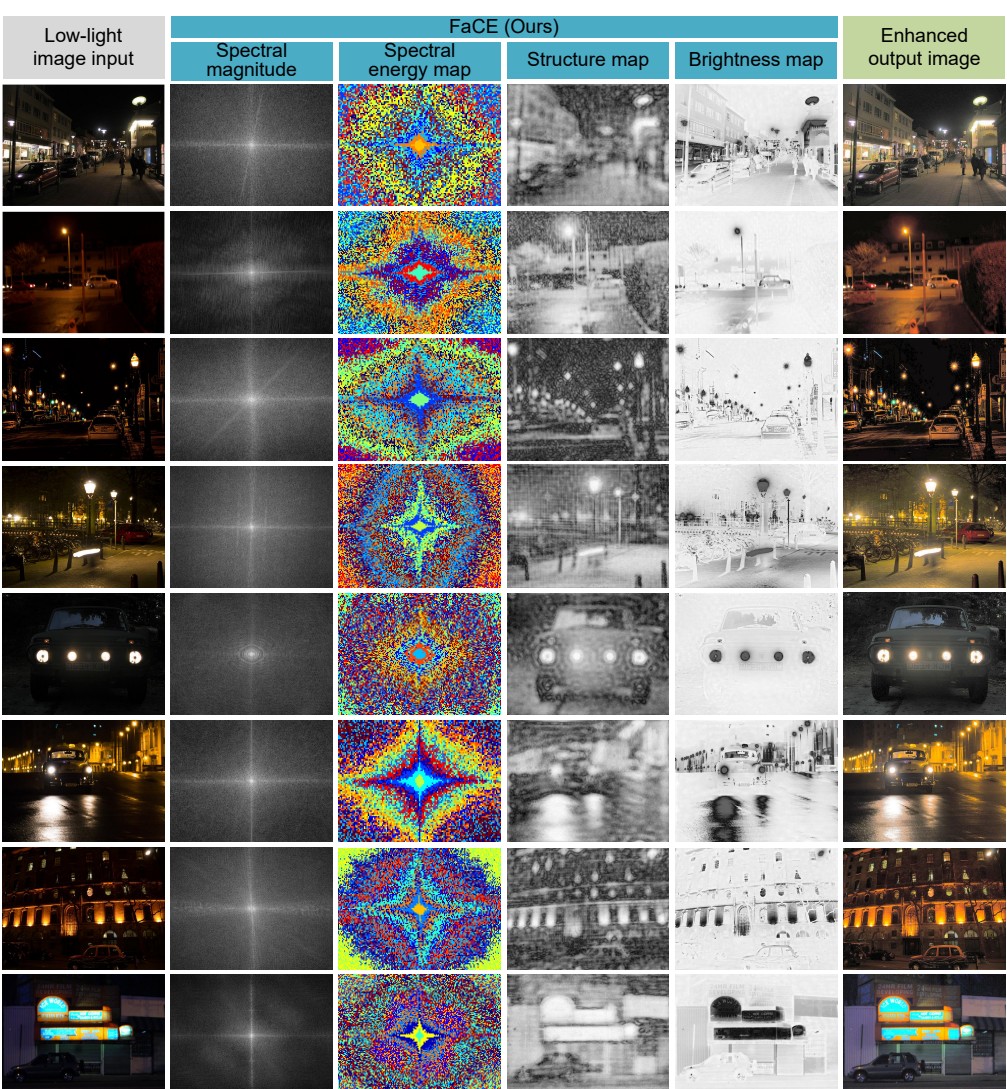

Figure 3: Visualization of FaCE on representative ExDark images. For each low-light input image (left), we show the Monogenic Fourier magnitude, the spectral energy map after frequency-aware clustering, the structure map and brightness map in the spatial domain, and the final enhanced output (right). These intermediate maps illustrate how our frequency-aware convex formulation brightens illumination while preserving scene structure.

Additional qualitative comparisons, exposure-level experiments on SICE, full DarkFace detection results, and complete ablation studies (including the monogenic transform, spectral clustering, and low-pass prior) are reported in the Appendix E.

Table 1 shows that FaCE, with 0 parameters and 0 training, achieves the best PSNR/SSIM among all 0P–0T baselines and approaches large supervised networks on both LOL-v1 and LOL-v2-Real. Table 2 shows that FaCE, despite being 0P–0T, attains the best perceptual quality (lowest PIQE) while keeping mAP and $AP_{50}$ competitive.

Figure 3 visualizes FaCE's frequency and spatial maps on ExDark: the spectral magnitude and energy maps show that illumination is concentrated in low-frequency bands, and the structure and brightness maps indicate that enhancement preserves scene structure. Appendix E (Figures E.5–E.7) further compares FaCE with LightenDiffusion, QuadPrior, DEnet, CLODE, and DarkIR on the Ex-

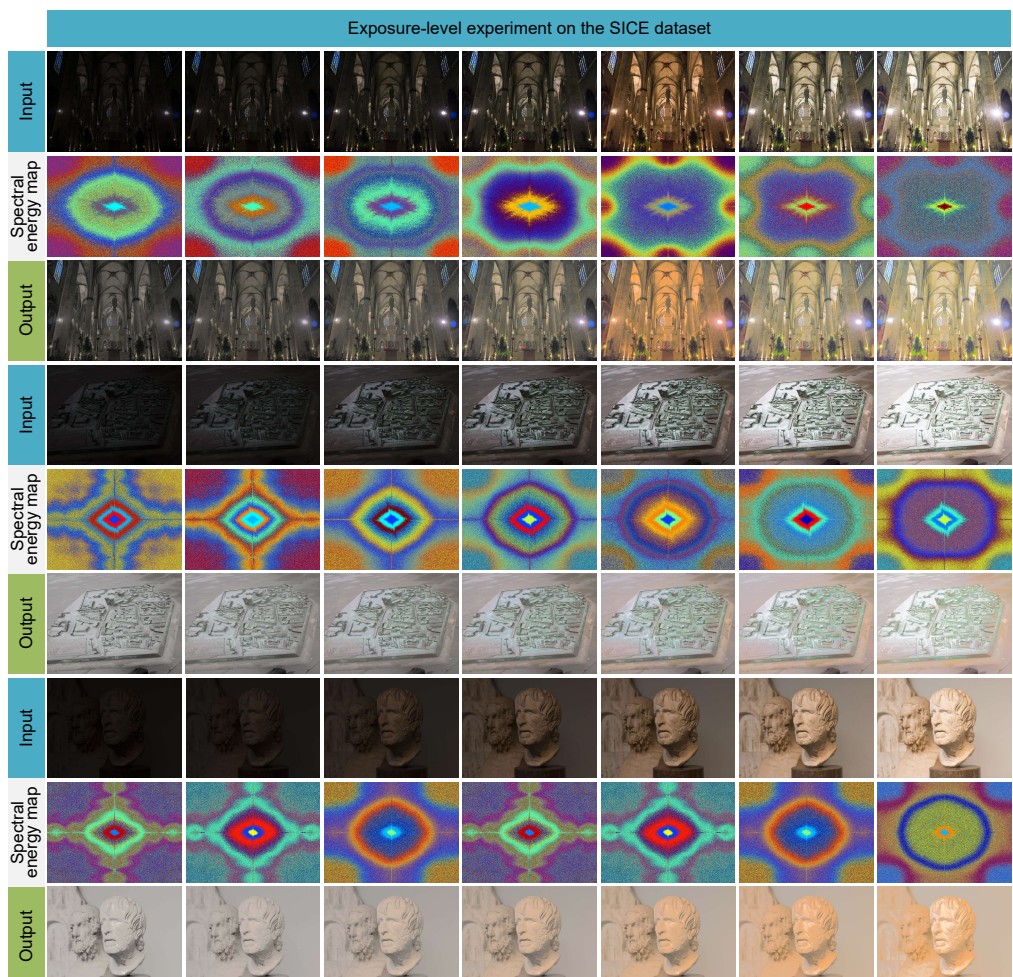

Figure 4: Exposure-level experiment on the SICE dataset. For each scene, we show input images with different exposure levels (top), the corresponding MFT spectral energy maps (middle), and FaCE outputs (bottom). FaCE uses the same 0P–0T configuration for all exposures and still produces stable, well-exposed results.

Dark people, car, and dog subsets. In these subsets, our 0P–0T method achieves visual quality comparable to strong learned baselines.

Figure 4 further shows on the SICE dataset that, with a fixed 0P–0T configuration, FaCE adapts smoothly to a wide range of exposure levels without re-training.

## 5 CONCLUSION

We cast LLIE as an MFT-grounded, frequency-domain inverse optimization and proposed FaCE—a training-free, illumination-only, strictly convex formulation with a spectral gain map from data-driven clustering. We proved existence, uniqueness, and stability, derived a discrete solver with no learned parameters, and observed consistent numerical stability and convergence.

Future work: we have not fully analyzed convergence rates, tight optimality conditions, or parameter sensitivity (e.g., $K, r_\tau, \lambda s$). Future work will develop principled self-calibration for frequency-domain clustering and priors (guiding $K$ and related parameters) and broaden evaluation against stronger learned priors.

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
