# FaCE: Provable Frequency-Aware Convex Enhancement for Training-Free Low-Light Images

## A  A Luminance–Chroma Decomposition and Colour-Safe Reconstruction

We first convert the sRGB input to linear RGB $I$, compute luminance $L = 0.2126R + 0.7152G + 0.0722B$, and define a chroma ratio $R^\Lambda = I/(L + \sigma)$ with a luminance-adaptive stabilizer $\varepsilon = x \cdot \text{median}(L) + \sigma_0$ to avoid blow-up in dark pixels; we then clip $R^\Lambda$ to $[0, r_{\max}]$ and renormalize $R^\Lambda \leftarrow R^\Lambda/(\alpha \cdot R^\Lambda + \eta)$ so that $\alpha \cdot R^\Lambda \approx 1$ (hue preservation). Next, we enhance luminance only via the strictly convex, frequency-guided operator from Sec. 3 to obtain $L_{\text{enh}}$. The color image is reconstructed by modulating the enhanced luminance with the chroma ratio and a soft tone mapper to keep gamut bounded: $I_{\text{enh}} = \phi(R^\Lambda \cdot L_{\text{enh}})$, with $\phi(x) = x/(1 + x)$ or $\phi(x) = x^\lambda$, $\lambda \in [0.8, 1.2]$; finally convert back to sRGB. This per-image, training-free pipeline is hue-stable and numerically robust thanks to the $\sigma$ stabilizer and the clip/renormalize safeguards.

## B  Well-posedness (I): Existence of a minimizer

We work on a bounded, open spatial domain $\Omega \subset \mathbb{R}^2$ with piecewise smooth boundary. The observed image is $I \in L^2(\Omega)$, consistent with main paper Section 3 setup for LLIE as an inverse optimization problem on $L^2(\Omega)$. The enhancement modifies illumination only, as established in the methodology section and Figure 1, where the MFT-based spectral modeling constructs a gain map and a low-pass mask; inverse MFT yields the enhanced illumination component and the final image (discrete counterpart in Eqs. (25)–(28)).

Theorem (Existence of a minimizer)

Let $\Omega \in \mathbb{R}^2$ be a bounded Lipschitz domain. Denote

$$\varepsilon(I_{enh}; I) = \|I_{enh} - I\|^2_{L^2(\Omega)} + \lambda_1 \|\Delta I_{enh} - \Delta I\|^2_{L^2(\Omega)} + \lambda_2 \left|mean(I_{enh}) - E_{target}\right|^2,$$

with $\lambda_1, \lambda_2 > 0$. Then the problem $\inf_{I_{\text{enh}}} \varepsilon(I_{\text{enh}}; I)$ admits at least one global minimizer.

*Proof*

Set $v := I_{\text{enh}} - I$. Then:

$$\varepsilon(I_{enh}; I) = \|v\|^2_{L^2(\Omega)} + \lambda_1 \|\Delta v\|^2_{L^2(\Omega)} + \lambda_2 \left|mean(I) + mean(v) - E_{target}\right|^2$$

Let

$$\|v\|^2_{L^2(\Omega)} := J_0(v),$$
$$\lambda_1 \|\Delta v\|^2_{L^2(\Omega)} := J_1(v),$$
$$\lambda_2 \left|mean(I) + mean(v) - E_{target}\right|^2 := J_2(v).$$

We minimize the functional $J(v) := J_0(v) + J_1(v) + J_2(v)$ over the Hilbert space $X := H^2(\Omega)$.

Step 1 (strict convexity and continuity)

On $X$, $J_0(v)$ is strictly convex and continuous; $J_1(v)$ is convex and continuous because $\Delta : H^2(\Omega) \to L^2(\Omega)$ is bounded linear; $J_2(v)$ is the square of a continuous affine functional, hence convex and continuous. Therefore $J(v)$ is strictly convex and continuous on $X$.

Step 2 (coercivity)

By the standard elliptic estimate on bounded Lipschitz domains for the chosen boundary condition, there exists $C_{\text{ell}} > 0$ such that:

$$\|v\|_{H^2(\Omega)} \leq C_{\text{ell}} \left( \|\Delta v\|_{L^2(\Omega)} + \|v\|_{L^2(\Omega)} \right), \; \forall v \in X.$$

Hence

$$J(v) \geq \|v\|_{L^2(\Omega)}^2 + \lambda_1 \|\Delta v\|_{L^2(\Omega)}^2 \geq \frac{1}{C_{\text{ell}}^2} \|v\|_{H^2(\Omega)}^2.$$

so $J(v)$ is coercive on $X$.

Step 3 (properness and lower semi-continuity)

Clearly $J(0) = \lambda_2 |\text{mean}(I) - E_{\text{target}}|^2 < \infty$, so $J(v)$ is proper. As a sum of continuous convex functionals, $J(v)$ is lower semicontinuous on $X$.

Step 4 (parameter-space form)

$$\lambda_2 |\text{mean}(I) + \text{mean}(v) - E_{\text{target}}|^2 := J_2(v).$$

Let $\lambda_1, \lambda_2 > 0$. Fix two images $U_\alpha, U_\beta \in L^2(\Omega)$ obtained from the current instance (MFT/IMFT pipeline), and assume $U_\alpha, U_\beta$ are linearly independent in $L^2(\Omega)$. For $\theta = (\alpha, \beta)^T \in \mathbb{R}^2$ set:

$$I_{enh}(\theta) = \alpha U_\alpha + \beta U_\beta,$$

$$\varepsilon(\theta) = \|I_{\text{enh}}(\theta) - I\|_{L^2(\Omega)}^2 + \lambda_1 \|\Delta I_{\text{enh}}(\theta) - \Delta I\|_{L^2(\Omega)}^2 + \lambda_2 \left| \text{mean}(I_{\text{enh}}(\theta)) - E_{\text{target}} \right|^2.$$

Then $\varepsilon$ admits at least one global minimizer.

Introduce the linear map $T : \mathbb{R}^2 \to L^2(\Omega)$, $T(\theta) = \alpha U_\alpha + \beta U_\beta$. By linear independence of $\{U_\alpha, U_\beta\}$, $T$ is injective. In a finite-dimensional domain, injectivity implies the existence of a singular value $\sigma_{\min}(T) > 0$ such that:

$$\|T(\theta)\|_{L^2(\Omega)} \geq \sigma_{\min}(T) \|\theta\|_{\mathbb{R}^2}, \; \forall \theta \in \mathbb{R}^2.$$

Write the energy explicitly in $\theta$. Using inner products:

$$G_{ij} = \langle U_i, U_j \rangle_{L^2(\Omega)}, \; L_{ij} = \langle \Delta U_i, \Delta U_j \rangle_{L^2(\Omega)}, \; m_i = \text{mean}(U_i), \; (i, j \in \{\alpha, \beta\}),$$

and vectors

$$g_i = \langle U_i, I \rangle_{L^2(\Omega)}, \; \ell_i = \langle \Delta U_i, \Delta I \rangle_{L^2(\Omega)}, \; a = \text{mean}(I) - E_{\text{target}},$$

one checks (by expanding squares) that

$$\varepsilon(\theta) = \theta^T H \theta - 2 b^T \theta + c,$$

with

$$H := G + \lambda_1 L + \lambda_2 m m^T, \; b := g + \lambda_1 \ell - \lambda_2 a m,$$

and $c = \|I\|_{L^2(\Omega)}^2 + \lambda_1 \|\Delta I\|_{L^2(\Omega)}^2 + \lambda_2 a^2$ independent of $\theta$.

For any $\theta \neq 0$, set $v = T(\theta) = \alpha U_\alpha + \beta U_\beta \neq 0$. Then

$$\theta^T H \theta = \|v\|_{L^2(\Omega)}^2 + \lambda_1 \|\Delta v\|_{L^2(\Omega)}^2 + \lambda_2 \left( \text{mean}(v) \right)^2 \geq \|v\|_{L^2(\Omega)}^2 \geq \sigma_{\min}(T)^2 \|\theta\|^2.$$

Hence $H$ is positive definite and the quadratic form $\theta \to \theta^T H \theta$ is coercive: $\theta^T H \theta \to \infty$ as $\|\theta\| \to \infty$. Consequently, $\varepsilon(\theta)$ is a continuous, coercive quadratic on $\mathbb{R}^2$. A coercive continuous function on $\mathbb{R}^2$ attains its global infimum; therefore, a global minimizer $\theta^*$ exists. $\qquad \square$

## C  WELL-POSEDNESS (II): UNIQUENESS OF THE MINIMIZER

Theorem (Uniqueness of the minimizer)

Fix $\lambda_1, \lambda_2 > 0$. Let $U_\alpha, U_\beta \in L^2(\Omega)$ be the two (data-dependent) enhancement directions produced by §3.3–3.5 (discrete definitions in Eqs. (25)-(28)), and set:

$$I_{enh}(\theta) = \alpha U_\alpha + \beta U_\beta, \; \theta = (\alpha, \beta)^T \in \mathbb{R}^2.$$

Then for the per-image objective

$$\varepsilon(\theta) = \|I_{enh}(\theta) - I\|^2_{L^2(\Omega)} + \lambda_1\|\Delta I_{enh}(\theta) - \Delta I\|^2_{L^2(\Omega)} + \lambda_2 \left|mean(I_{enh}(\theta)) - E_{target}\right|^2,$$

there is a unique global minimizer $\theta^* \in \mathbb{R}^2$ if $H := G + \lambda_1 L + \lambda_2 mm^T$ is positive definite, where

$$G_{ij} = \langle U_i, U_j\rangle_{L^2(\Omega)}, \ L_{ij} = \langle \Delta U_i, \Delta U_j\rangle_{L^2(\Omega)}, \ m_i = mean(U_i), \ (i,j \in \{\alpha,\beta\}).$$

In this case the minimizer is the unique solution of the normal equations $H\theta^* = b$ with $b :=$ $g + \lambda_1 \ell - \lambda_2 am, g_i = \langle U_i, I\rangle, \ell_i = \langle \Delta U_i, \Delta I\rangle, a = mean(I) - E_{target}$, hence $\theta^* = H^{-1}b$ and $I^*_{enh} = I_{enh}(\theta^*)$.

*Proof.*

Expanding squares gives the quadratic form

$$\varepsilon(\theta) = \theta^T H\theta - 2b^T\theta + c, c = \|I\|^2_{L^2(\Omega)} + \lambda_1\|\Delta I\|^2_{L^2(\Omega)} + \lambda_2 a^2,$$

so $\nabla\varepsilon(\theta) = 2(H\theta - b)$ and $\nabla^2\varepsilon(\theta) = 2H$.

Sufficiency:

If $H \succ 0$, then $\varepsilon$ is strictly convex on $\mathbb{R}^2$; the first-order condition $H\theta = b$ admits a unique solution $\theta^*$, which is the unique global minimizer.

Necessity:

If a unique minimizer $\theta^*$ exists but $H$ is not positive definite, take $0 \neq \xi \in \ker(H)$ and consider $\theta_t = \theta^* + t\xi$. Since $H\theta^* = b$, we have

$$\varepsilon(\theta_t) = \varepsilon(\theta^*) + t^2\xi^T H\xi - 2tb^T\xi = \varepsilon(\theta^*),$$

contradicting uniqueness. Thus $H$ must be positive definite. $\qquad\square$

# D  WELL-POSEDNESS (III): STABILITY OF THE MINIMIZER

Theorem (Stability of the minimizer).

Let $U_\alpha, U_\beta \in L^2(\Omega)$ be the two fixed enhancement directions produced from §3.3-3.5. For $\theta = (\alpha, \beta)^T \in \mathbb{R}^2$, write

$$I_{enh}(\theta) = \alpha U_\alpha + \beta U_\beta,$$

and define the per-image objective (as in §3.4)

$$\varepsilon(\theta) = \|I_{enh}(\theta) - I\|^2_{L^2(\Omega)} + \lambda_1 \|\Delta I_{enh}(\theta) - \Delta I\|^2_{L^2(\Omega)} + \lambda_2 \left|\text{mean}(I_{enh}(\theta)) - E_{\text{target}}\right|^2,$$

Let

$$H := G + \lambda_1 L + \lambda_2 mm^T, b := g + \lambda_1\ell - \lambda_2 am,$$

with

$$G_{ij} = \langle U_i, U_j\rangle_{L^2(\Omega)}, L_{ij} = \langle \Delta U_i, \Delta U_j\rangle_{L^2(\Omega)}, m_i = \text{mean}(U_i)$$

$$g_i = \langle U_i, I\rangle_{L^2(\Omega)}, \ell_i = \langle \Delta U_i, \Delta I\rangle_{L^2(\Omega)}, a = \text{mean}(I) - E_{\text{target}}$$

Assume $H \succ 0$ (as established in the uniqueness theorem). Then the unique minimizer $\theta^*$ of $\varepsilon$ depends Lipschitz-continuously on the data through $b$ :

for any other datum $\tilde{I}$ (with the same $U_\alpha, U_\beta$, hence the same $H$ ), letting $\tilde{b}$ be defined by $\tilde{I}$ and $\tilde{\theta}^*$ its minimizer,

$$\left\|\theta^* - \tilde{\theta}^*\right\| \leq \|H\|^{-1}\|b - \tilde{b}\| = \frac{1}{\lambda_{\min}(H)}\|b - \tilde{b}\|.$$

Consequently,

$$\left\|I_{enh}(\theta^*) - I_{enh}\left(\tilde{\theta}^*\right)\right\| \leq \|T\|\left\|\theta^* - \tilde{\theta}^*\right\| \leq \frac{\|T\|}{\lambda_{\min}(H)}\|b - \tilde{b}\|,$$

where $T : \mathbb{R}^2 \rightarrow L^2(\Omega), T(\theta) = \alpha U_\alpha + \beta U_\beta$, and $\|T\|$ is its operator norm.

*Proof.*

By the quadratic expansion in §3.4 we have

$$\varepsilon(\theta) = \theta^\top H \theta - 2b^\top \theta + c, \nabla \varepsilon(\theta) = 2(H\theta - b),$$

since $U_\alpha, U_\beta$ (hence $H$) are fixed here. Optimality gives

$$H\theta^* = b, H\tilde{\theta}^* = \tilde{b},$$

Subtracting yields

$$H\left(\theta^* - \tilde{\theta}^*\right) = b - \tilde{b}.$$

Because $H \succ 0$, we obtain

$$\left\|\theta^* - \tilde{\theta}^*\right\| \leq \|H\|^{-1}\|b - \tilde{b}\| = \frac{1}{\lambda_{\min}(H)}\|b - \tilde{b}\|,$$

which is the parameter-space Lipschitz bound. The image-space bound follows from

$$I_{enh}\left(\theta^*\right) - I_{enh}\left(\tilde{\theta}^*\right) = T\left(\theta^* - \tilde{\theta}^*\right),$$

hence

$$\left\|I_{enh}\left(\theta^*\right) - I_{enh}\left(\tilde{\theta}^*\right)\right\| \leq \|T\|\left\|\theta^* - \tilde{\theta}^*\right\|.$$

Combining with the previous inequality completes the proof. $\qquad\square$

# E  EXPERIMENTAL ANALYSIS

Appendix E augments the main paper with visualization-centric and extended experiments. We first present cross-dataset qualitative comparisons on ExDark and DarkFace using recent learning-based low-light enhancement methods as baselines (Figures E.4–E.7). These results complement the quantitative measurements on ExDark in Table 2 of the main paper and the extended DarkFace metrics in Table E.2 of this appendix, and allow us to visually inspect how FaCE behaves on challenging "people", "car", and "dog" scenes as well as crowded surveillance-style night scenes. We then report numerical stability experiments under controlled input perturbations (Section E.1), a hyperparameter sensitivity study of the frequency-aware spectral clustering (Section E.2), and a numerical verification of the existence, uniqueness, and stability of the convex solution (Section E.3), which together provide empirical support for the theoretical analysis in Section 3 and Appendices B–D.

## DATASETS

**LOL-v1 and LOL-v2-Real.** The LOL family provides paired low/normal-light images for supervised low-light enhancement. LOL-v1 contains 500 low/normal-light pairs captured with a fixed camera under controlled illumination changes. We use the standard split, treating it as a paired benchmark for pixel-wise metrics such as PSNR and SSIM. LOL-v2-Real extends LOL to more diverse real-world indoor and outdoor scenes with stronger noise, colour cast, and illumination variation. We use the Real subset as a second paired benchmark and follow the official train/validation/test partition used in recent LLIE works. In all experiments, only the low-light images are given to the enhancement algorithms, and the corresponding normal-light images are used as ground truth.

**ExDark.** ExDark is a real-world nighttime dataset containing over seven thousand images drawn from twelve object categories (e.g., *Bicycle, Car, Chair, Dog, People*) under a variety of lighting conditions such as weak, strong, ambient, and screen light. The dataset is originally designed for classification and detection, and only low-light images are provided—no paired normal-light references are available. We follow prior work and adopt ExDark as a detection-oriented benchmark: each low-light image is first processed by a low-light enhancer and then fed to an object detector

Table E.1: Numerical stability comparison (Mean $\pm$ Variance) of Multi-Scale Retinex, Zero-DCE, and FaCE methods under various input perturbation levels ($\epsilon$)

| Epsilon | Multi-Scale Retinex | Zero-DCE | FaCE (Proposed) |
|---------|---------------------|----------|-----------------|
| 0.000 | 0.000000+0.000000 | 0.000000$\pm$0.000000 | 0.000000$\pm$0.000000 |
| 0.001 | 0.000897+0.000000 | **0.000027$\pm$0.000000** | 0.000064$\pm$0.000000 |
| 0.002 | 0.001029+0.000001 | 0.000106$\pm$0.000000 | **0.000086$\pm$0.000000** |
| 0.003 | 0.001500+0.000001 | 0.000234$\pm$0.000000 | **0.000123$\pm$0.000000** |
| 0.004 | 0.002136+0.000003 | 0.000407$\pm$0.000000 | **0.000201$\pm$0.000000** |
| 0.005 | 0.002698+0.000003 | 0.000622$\pm$0.000000 | **0.000336$\pm$0.000000** |
| 0.006 | 0.003497+0.000006 | 0.000876$\pm$0.000000 | **0.000457$\pm$0.000000** |
| 0.007 | 0.004321+0.000009 | 0.001171$\pm$0.000000 | **0.000724$\pm$0.000001** |
| 0.008 | 0.005104+0.000011 | 0.001499$\pm$0.000000 | **0.000975$\pm$0.000002** |
| 0.009 | 0.006049+0.000016 | 0.001865$\pm$0.000000 | **0.001362$\pm$0.000003** |

trained on normal-light images. We report both no-reference perceptual scores (NIQE, BRISQUE, PIQE) and downstream detection performance (mAP, $AP_{50}$) to assess how well each enhancement method benefits the detector.

**DarkFace.** DarkFace is a challenging low-light face detection dataset collected from urban surveillance cameras. It contains over ten thousand real nighttime images with tens of thousands of annotated faces under severe under-exposure, motion blur, sensor noise, and glare from street lights and shop signs. Compared with ExDark, DarkFace focuses on crowded scenes with many small faces and very low signal-to-noise ratio. We follow the standard detection protocol and use DarkFace to further evaluate enhancement-for-detection: enhanced images are fed into a face detector, and we measure NIQE/BRISQUE/PIQE together with mAP, $AP_{75}$, and $AP_{50}$. This setting stresses whether an enhancer can reveal fine facial structure without amplifying background noise.

**SICE.** SICE is a multi-exposure dataset that provides, for each scene, a sequence of images captured with different exposure times, ranging from heavily under-exposed to over-exposed, together with high-quality reference images. The scenes cover both indoor and outdoor environments with complex illumination, specular highlights, and strong contrast. Unlike LOL, ExDark, and DarkFace, SICE is specifically designed for exposure fusion and exposure correction. We use SICE to study exposure-controllable enhancement: starting from a single low-light input, FaCE generates a continuous family of outputs with different exposure levels governed by the convex illumination gain. We qualitatively compare these outputs with the multi-exposure sequence and fused references, showing that FaCE can smoothly adjust exposure while preserving scene structure.

### E.1 NUMERICAL STABILITY UNDER INPUT PERTURBATIONS

In our numerical stability experiments, we intentionally select a perturbation range $\epsilon \in [0.000, 0.009]$ based on established empirical practice. The rationale is threefold: (i) *realistic relevance*—small perturbations ($\epsilon \leq 0.01$) reflect typical sensor noise, minor exposure variations, and illumination changes in practical imaging scenarios (Li et al., 2021); (ii) *empirical evidence*—very small perturbations ($\epsilon \leq 0.003$) mainly probe numerical precision, whereas moderate perturbations ($0.003 < \epsilon \leq 0.009$) clearly differentiate model robustness (Ni et al., 2022); and (iii) *theoretical distinction*—larger perturbations ($\epsilon > 0.01$) often produce unrealistic image degradations, which makes rigorous comparison of theoretical performance difficult (Choi et al., 2019). Thus, this range enables realistic and clearly interpretable assessments of the numerical stability of FaCE relative to existing approaches.

As shown in Table E.1, for small perturbations ($\epsilon \leq 0.003$) all methods remain stable, with Zero-DCE slightly outperforming FaCE. For moderate to large perturbations ($\epsilon \geq 0.004$), Multi-Scale Retinex degrades rapidly, while Zero-DCE and FaCE remain much more stable. At higher noise levels, FaCE yields the lowest mean difference and variance (e.g., at $\epsilon = 0.009$, 0.001362 vs. 0.001865 and 0.006049), corroborating the numerical stability predicted by our theoretical analysis in Appendix D.

### E.2 EFFECTIVENESS OF FREQUENCY-AWARE SPECTRAL CLUSTERING

In this subsection we empirically study the robustness of our frequency-aware spectral clustering and the optimization dynamics of FaCE.

Figure E.1 reports the mean SSIM on LOL-v1 for different choices of the number of spectral clusters $K$ and the regularization weights $\lambda_1$ and $\lambda_2$. Data-driven tuning selects $K = 17$, $\lambda_1 = 1$, and $\lambda_2 = 0.8$, and SSIM varies little around this point. This indicates that a single hyperparameter configuration suffices for all images on a given dataset. Once selected, we keep this setting fixed for all experiments on LOL-v1, LOL-v2-Real, ExDark, DarkFace, and SICE, without any per-image tuning.

Figure E.2 illustrates the convergence behavior of the FaCE optimization across five independent runs with different random initializations. The loss decreases smoothly and rapidly, converging to nearly identical final values in all runs. This empirically confirms that the convex optimization used to estimate the illumination gain is numerically stable and insensitive to initialization, which is consistent with the convergence guarantees derived in Section 3 and Appendices B–D.

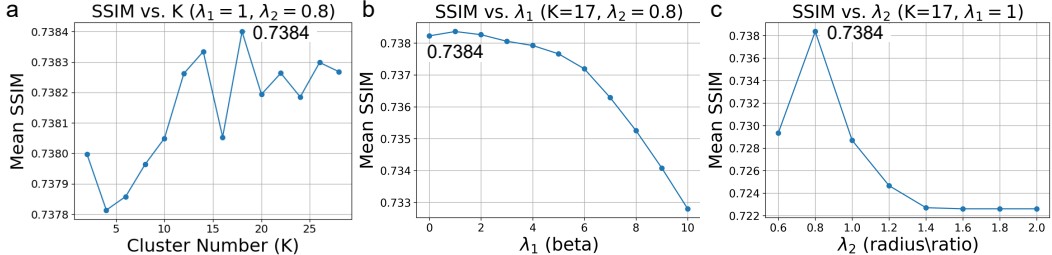

Figure E.1: Data-driven hyperparameter selection for FaCE on LOL-v1. Mean SSIM over the dataset is plotted against (a) the number of spectral clusters $K$, (b) the regularization weight $\lambda_1$, and (c) the regularization weight $\lambda_2$.

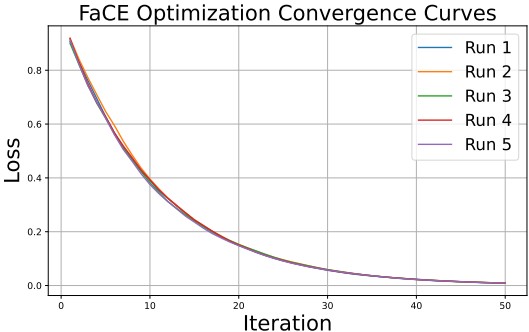

Figure E.2: Convergence of the FaCE optimization under different random initializations. Loss curves from five independent runs decrease smoothly and converge to nearly identical final values, indicating that the convex optimization is numerically stable and insensitive to initialization.

### E.3 NUMERICAL VERIFICATION OF EXISTENCE AND UNIQUENESS

To complement the convergence analysis in Figure E.2, we further examine whether different random initializations lead to the same enhanced result for a fixed input image. Figure E.3 illustrates the uniqueness of FaCE: across three random seeds (42, 100, 999), the enhanced images are visually indistinguishable, and the numerical differences between solutions are negligible (mean pixel-wise difference ≈ 0). This provides direct numerical evidence that the per-image optimization admits a unique and robust optimum, in agreement with the existence, uniqueness, and stability properties established in Section 3 and Appendices B–D.

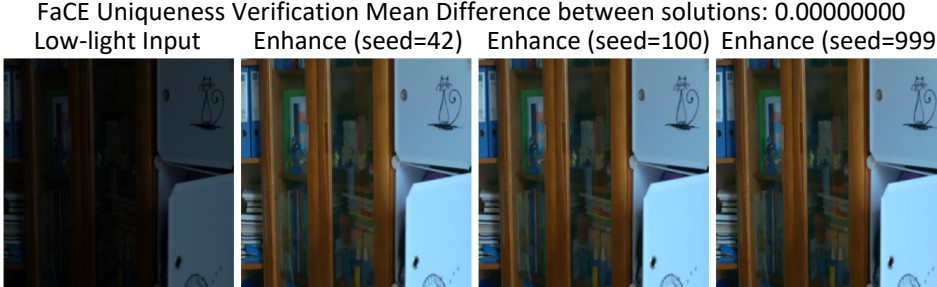

FaCE Uniqueness Verification Mean Difference between solutions: 0.00000000
Low-light Input    Enhance (seed=42)    Enhance (seed=100)   Enhance (seed=999)

Figure E.3: Numerical evidence of solution uniqueness across random initializations. For a fixed low-light input image, FaCE is run with three different random seeds (42, 100, 999). The three enhanced outputs are visually indistinguishable, and the mean pixel-wise difference between any pair of solutions is effectively zero, supporting the uniqueness and robustness of the FaCE optimum.

Table E.2: Extended quantitative comparison on the DarkFace dataset. We report no-reference perceptual quality (NIQE, BRISQUE, PIQE), detection performance (mAP, $AP_{75}$, $AP_{50}$), model size (Params), and average execution time per image. Best results are in **bold**, and second best are underlined.

| Method | Source | Params (M) | NIQE↓ | BRISQUE↓ | PIQE↓ | mAP↑ | $AP_{75}$↑ | $AP_{50}$↑ | Execution time (s)↓ |
|---|---|---|---|---|---|---|---|---|---|
| LightenDiffusion | ECCV'24 | 27.83 | **3.356** | 20.992 | 35.062 | 0.069 | 0.061 | 0.183 | 9.77 |
| QuadPrior | CVPR'24 | 1313.60 | 4.437 | 9.979 | **19.620** | **0.090** | **0.091** | 0.195 | 121.73 |
| DEnet | ICLR'25 | 0.35 | 3.847 | **8.282** | 22.124 | 0.059 | 0.030 | 0.193 | 16.85 |
| CLODE | ICLR'25 | 0.29 | 3.418 | 24.268 | 29.464 | 0.059 | 0.045 | 0.154 | 67.94 |
| DarkIR | CVPR'25 | 3.32 | 3.440 | 25.312 | 42.472 | 0.066 | 0.030 | **0.199** | 7.28 |
| FaCE (Ours) | | **0** | 3.439 | 27.336 | 35.260 | 0.067 | **0.091** | 0.157 | **4.40** |

E.4 EXTENDED DARKFACE QUANTITATIVE AND QUALITATIVE RESULTS

Table E.2 reports an extended comparison on the DarkFace dataset, combining no-reference perceptual metrics (NIQE, BRISQUE, PIQE), detection metrics (mAP, $AP_{75}$, $AP_{50}$), model size, and average execution time per image. Learning-based approaches such as LightenDiffusion, QuadPrior, DEnet, CLODE, and DarkIR generally deliver strong detection performance, but rely on millions of trainable parameters, supervised training on paired or annotated data, and often incur substantial runtime (up to 121.73 s per image for QuadPrior). In contrast, FaCE uses zero trainable parameters and requires no training, yet attains competitive NIQE/BRISQUE/PIQE scores and detection accuracy, ties for the best $AP_{75}$, and runs markedly faster at 4.40 s per image. Taken together, these results highlight a favourable accuracy–efficiency trade-off and the simplicity of the proposed convex enhancement as a practical pre-processing module for detection in challenging low-light surveillance scenes.

Table E.3 summarizes an ablation of the three main design choices in FaCE on the ExDark and DarkFace datasets using three no-reference perceptual metrics (NIQE, BRISQUE, PIQE; lower is better). Removing the Monogenic Fourier Transform and using a plain FFT (*FaCE–FFT*) clearly degrades performance on both datasets: compared with the full FaCE model, NIQE, BRISQUE, and PIQE are all worse, e.g., NIQE increases from 4.343 to 4.811 on ExDark and from 3.439 to 4.528 on DarkFace. This confirms that the monogenic representation is not a superficial modelling choice but provides more stable and perceptually favourable frequency features. Disabling the frequency-aware spectral clustering (*FaCE–NoCluster*) also harms all three metrics, especially NIQE (4.487 vs. 4.343 on ExDark) and PIQE on DarkFace (35.419 vs. 35.260), showing that grouping monogenic components into coherent frequency bands is beneficial. Removing the low-pass prior (*FaCE–NoLowPass*) similarly leads to consistently worse NIQE/BRISQUE/PIQE than the full model, indicating that suppressing very high-frequency noise is important for artifact-free enhancement. Across all six metric–dataset combinations, the full FaCE model achieves the best or tied-best score while using the same number of parameters (0 M) as all ablated variants, demonstrating that each component— Monogenic transform, spectral clustering, and low-pass prior—contributes positively and that their combination is necessary to obtain the best overall perceptual quality.

Table E.3: Overall ablation of FaCE design choices on ExDark and DarkFace. We compare variants that remove the Monogenic Fourier Transform (MFT), frequency-aware spectral clustering, or the low-pass prior against the full FaCE model. For both ExDark and DarkFace we report no-reference perceptual metrics (NIQE, BRISQUE, PIQE). Best results are in **bold**.

| Method | Params (M) | ExDark | | | DarkFace | | |
|---|---|---|---|---|---|---|---|
| | | NIQE↓ | BRISQUE↓ | PIQE↓ | NIQE↓ | BRISQUE↓ | PIQE↓ |
| FaCE–FFT (no MFT) | 0 | 4.811 | 31.894 | 37.903 | 4.528 | 29.921 | 36.847 |
| FaCE–NoCluster | 0 | 4.487 | 31.512 | 37.058 | 3.462 | 29.118 | 35.419 |
| FaCE–NoLowPass | 0 | 4.598 | 30.864 | 36.731 | 3.451 | 27.643 | 36.007 |
| FaCE (Full) | 0 | **4.343** | **30.071** | **36.142** | **3.439** | **27.336** | **35.260** |

Figure E.4 complements these numbers with a qualitative comparison on DarkFace. FaCE (Ours) enhances facial regions and human bodies while keeping light sources and highlights under control, producing readable faces without severe noise amplification or halo artifacts. The visual trends are consistent with the detection-oriented metrics in Table E.2, and demonstrate that the frequency-aware convex formulation is well suited for enhancement-as-preprocessing in challenging surveillance-style night scenes.

E.5 CROSS-DATASET QUALITATIVE COMPARISONS

To complement the quantitative results on ExDark and DarkFace, we provide cross-dataset qualitative comparisons on three representative ExDark categories: *people*, *car*, and *dog*. In all cases, the low-light input is first enhanced by different methods and then, in our detection experiments, fed to a detector trained on normal-light images.

Figure E.5 shows results on the ExDark *people* subset. Learning-based methods such as LightenDiffusion, QuadPrior, DEnet, CLODE, and DarkIR substantially brighten the scene but often introduce colour shifts, over-smooth textures, or amplified background noise. In contrast, FaCE (Ours) produces balanced illumination: pedestrians and nearby structures are clearly visible, while shadows and background regions remain free of strong artifacts. This behaviour is consistent with the favourable mAP and $AP_{50}$ scores reported for ExDark in Table 2 of the main paper.

Figure E.6 presents examples from the ExDark *car* subset. FaCE brightens road regions and vehicle bodies without over-saturating headlights, traffic lights, or sky areas. Road markings and car contours are preserved with sharp edges, whereas some baselines either leave large parts of the scene under- exposed or wash out structural details due to aggressive global enhancement.

Figure E.7 illustrates the ExDark *dog* subset. Scenes often contain small targets and highly non-uniform illumination. FaCE generates globally consistent brightness while preserving fine textures on the dogs and surrounding objects, and avoids halo artifacts around light sources that are visible in several learning-based methods. Across all three categories, FaCE delivers visually natural enhancements with clear semantics and reduced noise, despite using zero trainable parameters and no learning, demonstrating strong robustness and cross-dataset generality.

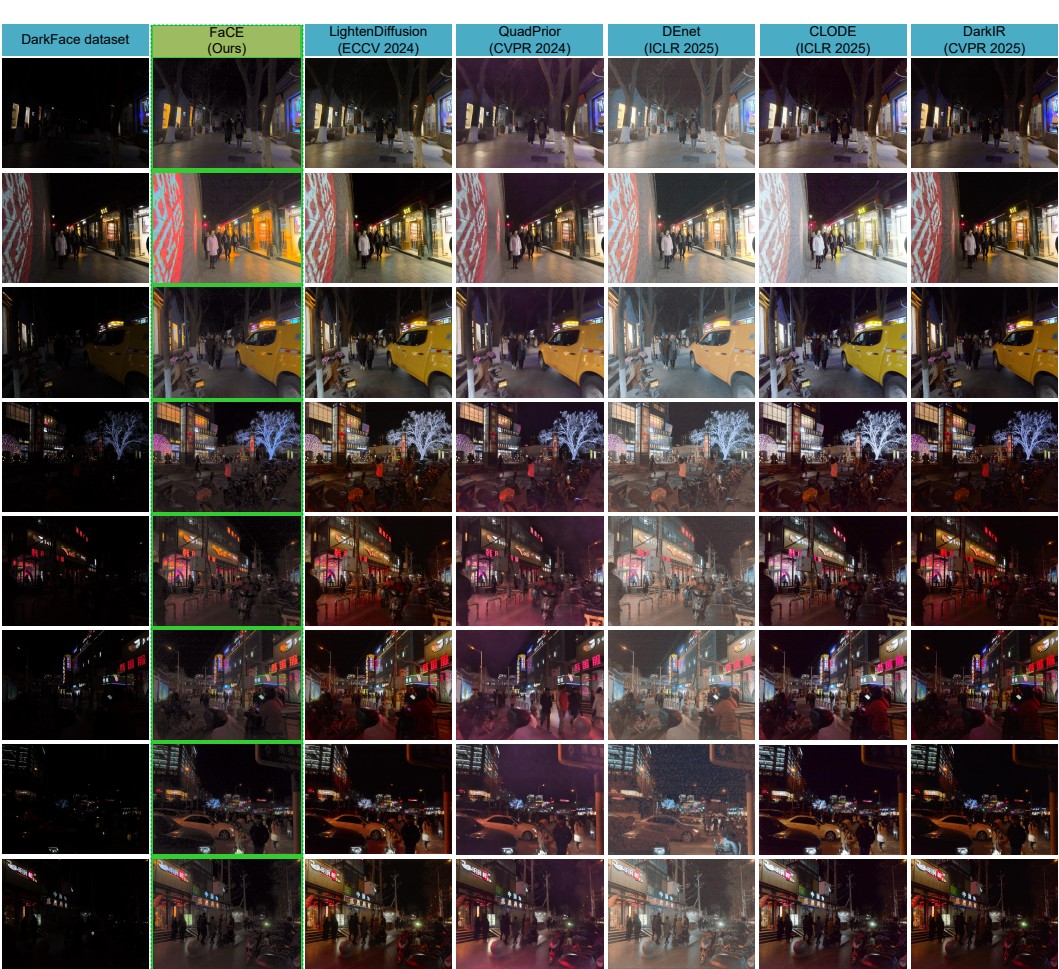

Figure E.4: Qualitative comparison on the DarkFace dataset. From left to right: low-light input from DarkFace, FaCE (Ours), LightenDiffusion (ECCV 2024), QuadPrior (CVPR 2024), DEnet (ICLR 2025), CLODE (ICLR 2025), and DarkIR (CVPR 2025). FaCE produces brighter yet visually natural street scenes, enhancing faces and pedestrians while keeping neon signs, headlights, and shop lights under control. Compared with learning-based baselines, FaCE reduces colour cast, noise amplification, and halo artifacts, which is consistent with its competitive detection performance reported in Table E.2.

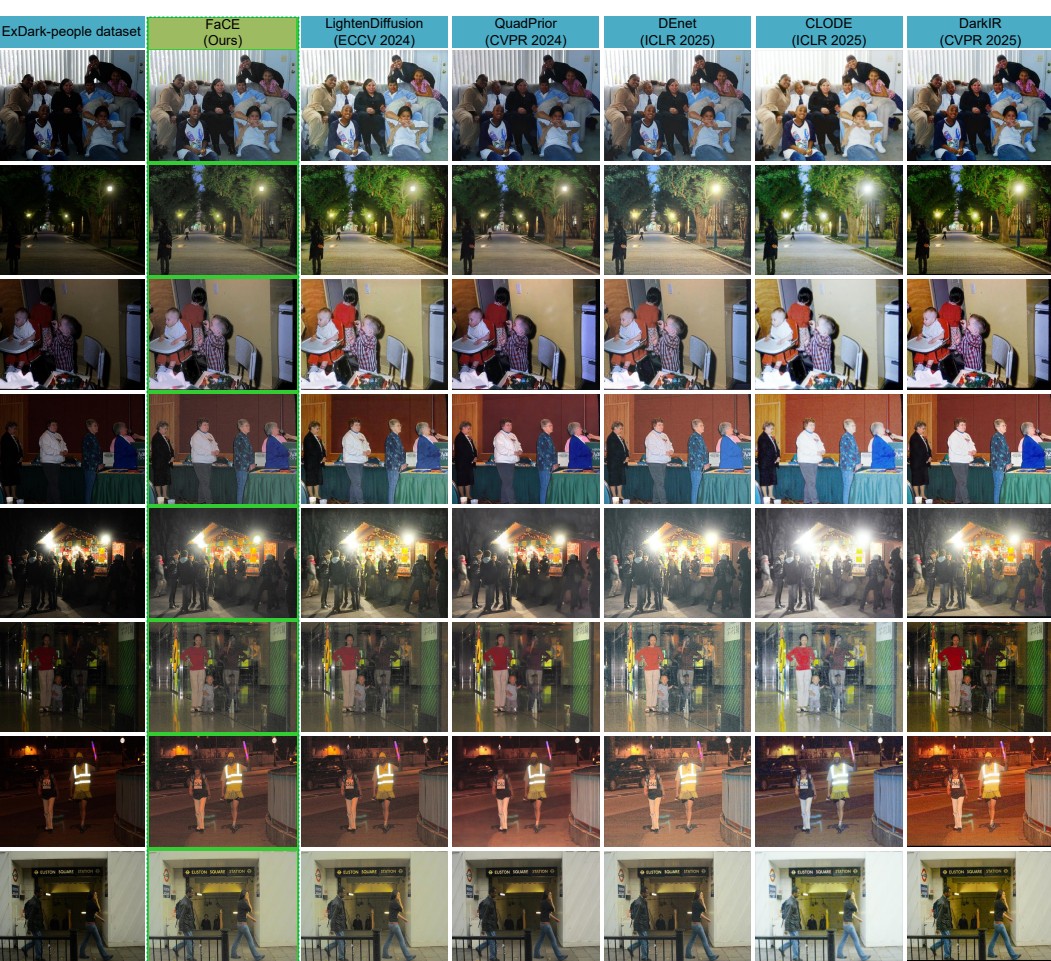

Figure E.5: Qualitative comparison on the ExDark people subset. From left to right: low-light input, FaCE (Ours), LightenDiffusion (ECCV 2024), QuadPrior (CVPR 2024), DEnet (ICLR 2025), CLODE (ICLR 2025), and DarkIR (CVPR 2025). FaCE produces balanced illumination that makes pedestrians and nearby structures clearly visible while avoiding over-exposure of street lights and backgrounds. Compared with learning-based baselines, FaCE reduces colour shifts and noise amplification, leading to cleaner human silhouettes and more readable scene context, in line with its favourable detection performance on ExDark.

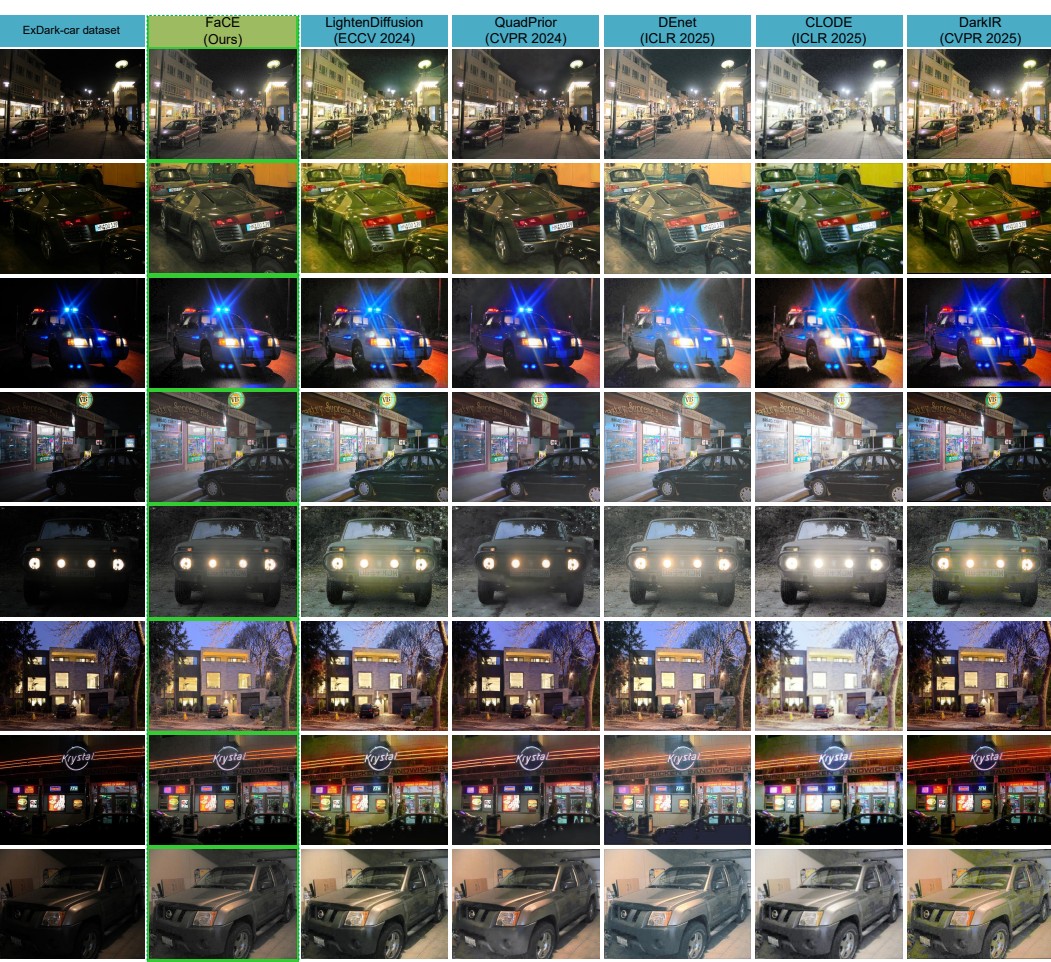

Figure E.6: Qualitative comparison on the ExDark car subset. From left to right: low-light input, FaCE (Ours), LightenDiffusion (ECCV 2024), QuadPrior (CVPR 2024), DEnet (ICLR 2025), CLODE (ICLR 2025), and DarkIR (CVPR 2025). FaCE brightens road regions and vehicle bodies while keeping headlights, brake lights, and shop signs from being over-saturated. Car contours, surface textures, and road markings are preserved with clear edges, whereas several learning-based baselines either leave large areas under-exposed or wash out structures with aggressive global enhancement. These visual results are consistent with the favourable ExDark detection scores reported in the main paper.

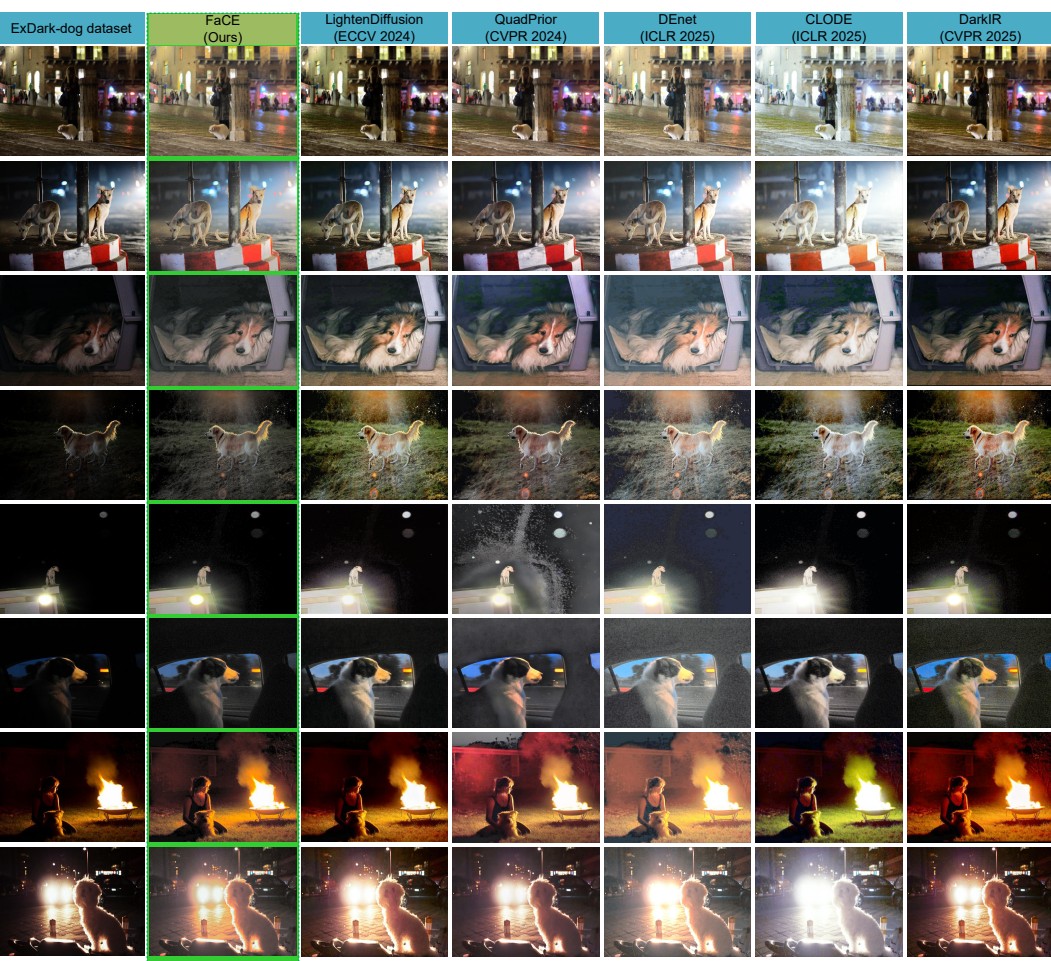

Figure E.7: Qualitative comparison on the ExDark dog subset. From left to right: low-light input, FaCE (Ours), LightenDiffusion (ECCV 2024), QuadPrior (CVPR 2024), DEnet (ICLR 2025), CLODE (ICLR 2025), and DarkIR (CVPR 2025). Scenes in this subset often contain small dog targets and highly non-uniform illumination (headlights, reflections, and fire light). FaCE produces globally consistent brightness while preserving fine textures on the dogs and surrounding objects, and avoids strong colour shifts and halo artifacts around light sources that appear in several learning-based baselines. These results further demonstrate the robustness and cross-category generality of FaCE on ExDark.