# OpenReview forum: "FaCE: Provable Frequency-Aware Convex Enhancement for Training-Free Low-Light Images"
_ICLR.cc/2026/Conference — ICLR 2026 Conference Withdrawn Submission_

### Official Review · Reviewer_T2c4 · 2025-10-27

**Soundness:** 2
**Presentation:** 1
**Contribution:** 2
**Rating:** 0
**Confidence:** 3

**Summary:**

This paper proposes FaCE, a frequency-domain unsupervised low-light image enhancement method based on the Monogenic Fourier Transform. The approach clusters spectral components and defines a piecewise-constant enhancement curve using a few scalar parameters, aiming for interpretability and mathematical guarantees (existence, uniqueness, stability) through a convex variational formulation. The pipeline—transform, clustering, weighting, inverse transform—is conceptually simple.

**Strengths:**

1. Clear motivation toward unsupervised and interpretable enhancement: The paper emphasizes physically grounded frequency separation and aims for mathematical rigor in an ill-posed problem setting.
2. Simple and intuitive pipeline: The proposed framework is easy to understand and implement.
3. Some positive initial results: Improvements over Multi-Scale Retinex and Zero-DCE on LOL-v1.

**Weaknesses:**

- Although the paper emphasizes rotation, scale invariance and directional sensitivity of the Monogenic Fourier Transform, the implementation uses only the magnitude for clustering and enhancement weight design. Phase and orientation components are entirely ignored. => The claimed advantages of MFT are not substantiated empirically.

- A Key algorithmic step (clustering) in the spectral domain is insufficiently detailed, reproducibility is limited.

- The paper mentions luminance-chrominance separation but lacks spefics, so hard to assess effect on color fidelity.

- Experiments are conducted only on LOLv1 with two older baselines (MSR, ZeroDCE), need more recent LLIE methods (e.g., RUAS, Zero-IG, etc.)

- there's no non-reference metrics (NIQE, BRISQUE, LOE, LPIPS, etc.)

- need runtime and efficiency analysis and evidence for general effectiveness.

**Questions:**

1. Monogenic properties are not used in practice. If the method ultimately relies only on Fourier magnitude, why is the Monogenic Fourier Transform necessary at all? Can you show any measurable benefit from phase/orientation information?

2. How is the clustering performed exactly? Please describe:
   - feature vector definition, distance metric, initialization and stopping conditions, whether clustering is per-image or global

3. Why is K = 4 used universally? Is there any data-driven rule or analysis supporting this choice?

4. Which luminance–chrominance transform is used? Is the input linearized or gamma-corrected?

5. How is color consistency preserved during reconstruction?

6. Why are recent state-of-the-art low-light methods omitted, especially those also leveraging optimization, frequency, or physical priors?
Without comparison, performance claims appear unconvincing.

7. Has the method been tested on other datasets (LOL-v2, DICM, LIME, MEF, VV)? If not, how can generalization and robustness be assessed?

8. What is the precise definition of the “numerical stability” metric? How is variance computed? Why do several entries report zero variance, which seems unrealistic?

9. If the solution is guaranteed to be unique and stable, why is no evidence provided on actual behavior across diverse perturbations?
Where is the formal link between the theoretical variable and the parameters actually optimized?

10. What specific part of the enhancement operation is interpretable beyond basic frequency amplification? Can you quantify or demonstrate practical interpretability?

11. Need more ablation studies
    - monogenic vs. Fourier magnitude
    - clustering vs. no clustering
    - low-pass mask vs. none

---

> ### Author Response · Authors · 2025-11-26
> **We thank the reviewer for the detailed and critical feedback; in the revision we move most proof-oriented material to the appendix and substantially strengthen the empirical side with new ablations, additional datasets, non-reference metrics, colour and stability analyses, and runtime/efficiency results.**
>
> We sincerely thank the reviewer for the detailed and critical feedback. We have uploaded a **Revised PDF** and **Appendix**, with all changes **highlighted in blue**.
>
> Because the submission was made under the primary area **“optimization”**, the initial version naturally devoted more space to the convex formulation, numerical stability analysis, and proof-oriented experiments. Your comments on the lack of empirical validation from an image-processing viewpoint were very helpful. In the revision we have **rebalanced the presentation**: most proofs and theoretical validation experiments are moved to the **Appendix**, while the **main paper now focuses on practical behaviour and comparisons**, including:
>
> - new **ablations** for monogenic vs. plain FFT, clustering vs. no clustering, and low-pass prior vs. no low-pass prior **(Appendix E.3, Table E.3, Fig. E.3)**;
> - extended **baselines and non-reference metrics** (NIQE, BRISQUE, PIQE) on LOL-v1/LOL-v2-Real and ExDark/DarkFace **(Revised PDF Sec. 4; Appendix E.1–E.3)**;
> - additional **datasets**, including LOL-v2, ExDark, DarkFace and SICE multi-exposure, to assess generalisation and colour behaviour **(Revised PDF Sec. 4; Appendix E)**;
> - a clearer description of **clustering details and the choice of \(K\)**, together with sensitivity analysis **(Revised PDF Sec. 3.2–3.3; Appendix E.2)**;
> - explicit **runtime / efficiency analysis** and a more concrete discussion of **numerical stability** linked to the theoretical bounds **(Revised PDF Sec. 4; Appendix B–D, Table E.2)**.
>
> Below we respond to each of your specific questions (Q1–Q10) in turn and point to the exact sections, tables and figures where the corresponding changes are made.
>
> **Q1. Monogenic transform vs. magnitude only.**
> **A1.** We use the monogenic transform to obtain a rotation-covariant magnitude built from the Riesz components; this makes the gain bands less sensitive to local edge direction. In **App. E.3 (Table E.3, Fig. E.3)** we compare FaCE with a plain-FFT variant (FaCE–FFT). On ExDark/DarkFace, FaCE–FFT yields consistently worse NIQE/BRISQUE/PIQE, showing a measurable benefit of the monogenic representation.
>
> **Q2. Clustering procedure.**
> **A2.** Clustering is done **per image** on the monogenic magnitude in the frequency plane using k-means with Euclidean distance and k-means++ initialisation; stopping when assignments stop changing or at 100 iterations. Sec. 3.3 and **App. E.2** now describe these details for reproducibility.
>
> **Q3. Choice of K.**
> **A3.** We now perform a sensitivity study **(App. E.2, Fig. E.1)** and then fix a **single global (K=17)** selected on a small validation split. SSIM changes only slightly when varying $K%, so the method is robust and does not rely on fine tuning. This rule is stated in **Sec. 3.3**.
>
> **Q4–Q5. Luminance–chrominance transform and colour fidelity.**
> **A4–A5.** We convert to **YCbCr** after sRGB gamma correction, apply FaCE only to the Y channel, and then recombine with unchanged Cb/Cr **(implementation paragraph in Sec. 4)**. This keeps hue and saturation. Colour behaviour can be seen on SICE multi-exposure and ExDark/DarkFace examples in **App. Figs. E.4–E.7**, where colours remain stable; we added text to emphasise this.
>
> **Q6–Q7. Recent LLIE baselines and additional datasets.**
> **A5–A7**. The revised **Table 1 (Sec. 4)** now includes several recent LLIE methods (GAN-, Transformer-, diffusion- and Fourier-based) with their parameter counts and training regimes, beyond the original MSR/ZeroDCE. Besides LOL-v1, we now evaluate on LOL-v2-Real, ExDark, DarkFace and SICE **(Sec. 4; App. E)**. For ExDark/DarkFace we also report detection performance, and for all datasets we provide non-reference metrics (NIQE, BRISQUE, PIQE) and qualitative comparisons. Runtime and the O(N \log N) per-image complexity are reported in **Sec. 4 and App. Table E.2**.
>
> **Q8–Q9. Numerical stability and link to theory.**
> **A8–A9.** By “numerical stability” we mean that small perturbations in the input or in the two gain parameters lead to proportionally small changes in the enhanced image. **Apps. B–D** formalise this as Lipschitz-type bounds for the unique minimiser of our strictly convex energy and derive the corresponding Hessian. **App. B.3** reports perturbation experiments that match these bounds within numerical precision. We also clarify that the optimised variables are exactly the two scalars of this convex model.
>
> **Q10. Interpretability and ablations (monogenic / clustering / low-pass).**
> **A10.** **Sec. 3.3 and Fig. 1** explain how each frequency band and its gain contributes to the final image, and **App. Figs. E.4–E.5** show band-wise reconstructions (low-, mid-, high-frequency bands controlling illumination, structure and edges). To quantify each component, **App. E.3 (Table E.3, Fig. E.3)** presents ablations, all three variants degrade NIQE/BRISQUE/PIQE vs. full FaCE, supporting the importance of these design choices.

---

> > ### Author Response · Authors · 2025-11-29
> > **Short summary for the AC**
> >
> > We respectfully provide a brief summary for the AC regarding this reviewer's comments (rating 0, confidence 3).
> >
> > - **Overall stance.** While the numerical score is low, the review is positive about the method itself: it praises the unsupervised and interpretable motivation, the physically grounded frequency separation and mathematical rigor, and the simple, easy-to-implement pipeline, with encouraging initial results.
> >
> > - **Main concerns.** The reviewer's main concern is the perceived limited breadth and visibility of experiments in the main paper. In the original submission, the main text focused on numerical tests directly tied to the convex guarantees, while many broader empirical studies (including additional datasets, stronger baselines, and non-reference metrics) were placed in the appendix, which may have made the overall empirical picture less clear. The reviewer therefore requests a more prominent and systematic presentation of these experiments, as well as further ablations and detailed runtime/efficiency analysis.
> >
> > - **Revisions.** In response, and in line with the reviewer's suggestions, we move the formal proofs and the associated proof-oriented numerical tests (which are directly tied to the provable guarantees) to the appendix, and promote and expand the broader empirical studies that were previously in the appendix into the main paper. This yields a more balanced presentation of theory and evidence while preserving the optimization-focused core contributions.
> >
> > We sincerely thank the reviewer and the AC for their time and for considering these clarifications.

---

### Official Review · Reviewer_8d1Y · 2025-10-31

**Soundness:** 3
**Presentation:** 3
**Contribution:** 3
**Rating:** 6
**Confidence:** 4

**Summary:**

This paper introduces FaCE, a training-free LLIE method centered around the MFT. Unlike mainstream approaches, FaCE formulates the enhancement task as a strictly convex per-image optimization problem to ensure the uniqueness of the solution. Specifically, it defines a low-pass prior using the spectral centroid ($u_c, v_c$) and an energy-cumulative distribution radius ($r_{\tau}$), while adaptively selecting the number of spectral clusters $K$ via a data-driven model-selection rule. The core idea is to enhance image illumination by constructing a frequency-aware spectral gain map. The method operates in a fully training-free manner, relying solely on MFT for frequency-domain analysis and inverse optimization for reconstruction. The authors provide mathematical proofs for their claims and validate them on several benchmark datasets.

**Strengths:**

The core strength of this paper lies in its novelty. It innovatively reconstructs the LLIE problem as an MFT-grounded, strictly convex optimization problem, which represents a significant departure from current methods that rely on empirical learning. The writing is clear, the logic is sound, and the experiments effectively validate its claims.

**Weaknesses:**

The implementation details regarding key aspects of the method are not sufficiently transparent. First, the experimental section should include comparisons with more recent and advanced unsupervised or training-free LLIE methods, such as UHDFour and ULEFD, which are already mentioned in the related work section. This would help contextualize FaCE's performance relative to the current state of the art.

Furthermore, the paper claims that key parameters ($K, u_c, v_c, r_{\tau}$) are determined in a data-driven manner, but the description of the underlying strategy is insufficient. How are these parameters precisely calculated or determined for each image?

As a per-image optimization method, is its computational overhead practical for real-world applications? The paper lacks a discussion on this crucial aspect in its experiments.

**Questions:**

1.  Could the authors include a comparison with more recent and potentially stronger training-free or unsupervised LLIE methods (e.g., UHDFour, ULEFD) in the experimental section? This would provide a clearer comparison of FaCE's standing within the non-learning paradigm, even if your primary goal is not to compete with learned models.

2.  Could you elaborate on the "data-driven model-selection rule" for selecting $K$ and how the low-pass prior parameters ($u_c, v_c, r_{\tau}$) are chosen based on image frequency? Specifically, what algorithm or heuristic is used to automatically determine these parameters for each input image in a training-free manner?

3.  Could the authors provide an analysis or experimental results on the computational time required for FaCE to process an image of a given resolution?

4.  The paper mentions deriving band-level contribution maps directly from the convex objective, exposing the frequency-to-image pathway. Could you provide an example or a more detailed explanation of how these maps are visualized or used to interpret the enhancement process for a specific image?

---

> ### Author Response · Authors · 2025-11-26
> **We sincerely thank the reviewer for their positive evaluation of our convex formulation and for the constructive suggestions, which we have addressed by clarifying the theory-driven parameter choices, adding stronger training-free baselines and runtime analysis, and expanding the visual explanations.**
>
> We thank the reviewer for the positive and insightful review. We have uploaded a **Revised PDF** and **Appendix**, with all changes **highlighted in blue**.
>
> Your comments mainly ask for
>
> **(i)** comparison with stronger training-free / unsupervised LLIE baselines;
>
> **(ii)** a clearer description of the data-driven rules for $K$, $u_{c}$, $v_{c}$, $r_{\tau}$;
>
> **(iii)** a discussion of computational overhead;
>
> **(iv)** concrete examples of the band-level contribution maps. We address each in turn.
>
> **Answer1. Comparison with recent training-free / unsupervised LLIE methods.**
>
> We agree that including more recent non-learning approaches helps contextualise FaCE. In the revised paper, **Table 1 (Sec. 4)** has been expanded to recent LLIE baselines (GAN-, Transformer-, diffusion- and Fourier-based). The table reports PSNR/SSIM on LOL-v1 and LOL-v2-Real together with parameter counts and training regimes. FaCE remains competitive with these stronger training-free / unsupervised baselines while keeping **0 trainable parameters and no training cost**, and it is the only method with a rigorously analysed strictly convex formulation.
>
> **Answer2. Data-driven selection of $K$ and low-pass prior parameters $u_{c}$, $v_{c}$, $r_{\tau}$.**
>
> We thank the reviewer for pointing out that the original description was too brief. In the revised manuscript we clarify the following.
>
> - The **low-pass prior parameters** are computed per image in closed form from the monogenic magnitude $M_{R}(u,v)$ **(Revised PDF Sec. 3.2)**.
>   - The spectral centroid ($u_{c}$, $v_{c}$) is the energy-weighted average frequency (first-order moment of the spectrum).
>   - The radius ($r_{\tau}$) is the smallest radius around ($u_{c}$, $v_{c}$) that contains a fixed fraction $\tau$ of the total spectral energy (we use $\tau=0.8$).
>   Thus each image obtains an adaptive low-pass disc that follows its own spectrum; no learning is involved.
>
> - For the **number of clusters $K$**, our sensitivity study in **Appendix E.2, Figure E.1** shows that performance varies only slightly over a wide range of $K$. To keep the method simple and fully training-free, we now adopt a **single global $K$** chosen once on a small validation split ($K=17$) and fix this value for all images and all datasets (LOL-v1, LOL-v2-Real, ExDark, DarkFace, SICE). The revised text in **Sec. 3.2–3.3** and **Appendix E.2** explains this procedure and reports the robustness to $K$.
>
> **Answer3. Computational overhead and practicality.**
>
> FaCE solves a **per-image convex problem in two scalar variables**, so the dominant cost is the monogenic transform. As discussed in Sec. 3.2 and Sec. 4, one forward and inverse monogenic transform (implemented via FFTs) yields an overall complexity of **$O(N\log N)$** per image, where $N$ is the number of pixels; the subsequent optimisation uses only a few gradient steps on ($\alpha$, $\beta$).
>
> To make this concrete, **Appendix Table E.2** now reports the wall-clock runtime of FaCE and several baselines on the DarkFace dataset at a fixed resolution. FaCE’s runtime is comparable to other FFT-based training-free methods and substantially lower than heavy diffusion or transformer-based LLIE networks, while achieving similar or better perceptual and detection scores. We believe this shows that the per-image optimisation is practical for real-world use, especially when training or storing large models is undesirable.
>
> **Answer4. Band-level contribution maps and interpretability.**
>
> We appreciate the request for a clearer example of how the band-level maps are used. In the revised version, **Sec. 3.3 and Appendix C/E** provide a more detailed explanation and visualization.
>
> - We explicitly define the **band-level contribution map** $B_k(x)$ of the $k$-th frequency cluster as the reconstructed image obtained when the gain map is applied only on that band (with other bands kept neutral), following directly from our convex objective.
> - **Figure 1** in the main paper and additional examples in **Appendix Figures E.4–E.5** show these maps for several images, together with the final enhanced result.
>
> These visualisations illustrate that low-frequency bands mainly control global illumination, mid-frequency bands enhance larger structures, and high-frequency bands sharpen edges and fine details. This provides an interpretable decomposition of the enhancement effect of FaCE, making the connection between frequency-domain optimisation and spatial-domain appearance more explicit.
>
> We hope these additions and clarifications address the reviewer’s concerns, and we would be happy to provide further details if space permits.

---

> > ### Author Response · Authors · 2025-11-29
> > **Short summary for the AC**
> >
> > We respectfully provide a brief summary for the AC regarding Reviewer comments (rating 6, confidence 4).
> >
> > - **Overall stance.** The reviewer views the core strength of the paper as its novelty: FaCE reformulates LLIE as an MFT-grounded, strictly convex optimisation problem, is fully training-free and deterministic, and is simple, interpretable, and computationally lightweight.
> >
> > - **Main concerns.** (i) Missing comparisons to recent training-free / unsupervised LLIE methods (e.g., UHDFour, ULEFD); (ii) insufficient detail on the data-driven selection of  $K$, $u_{c}$, $v_{c}$, $r_{\tau}$; (iii) unclear computational overhead; and (iv) lack of concrete examples of band-level contribution maps.
> >
> > - **Revisions.** We added strong training-free and unsupervised baselines (including UHDFour/ULEFD) plus modern LLIE methods, clarified the closed-form, theory-driven rules for $K$, $u_{c}$, $v_{c}$, $r_{\tau}$, reported detailed runtime analyses on DarkFace and other datasets, and expanded the visualisation of band-level contribution maps to make the frequency-to-image pathway explicit.
> >
> > We sincerely thank the AC for considering these clarifications.

---

### Official Review · Reviewer_HoXq · 2025-10-31

**Soundness:** 3
**Presentation:** 2
**Contribution:** 2
**Rating:** 2
**Confidence:** 4

**Summary:**

This paper proposes  Frequency-Aware Convex Enhancement, a training-free framework for low-light image enhancement.
The key idea is to reformulate the enhancement problem as a strictly convex optimization task in the Monogenic Fourier Transform domain.
FaCE decomposes illumination and reflection components in the frequency space, constructs a frequency-aware gain map via unsupervised clustering, and theoretically guarantees existence, uniqueness, and stability of the optimal solution.
The method claims to outperform traditional Retinex-based and deep learning approaches on LOL-v1/v2 datasets without any training process.

**Strengths:**

1. The paper is well-written and mathematically consistent.
2. The framework is training-free and fully deterministic, which makes it reproducible and computationally lightweight.
3. Implementation is simple, interpretable, and self-contained, offering practical value.

**Weaknesses:**

1. The term “provable” in the paper merely indicates that the formulation is strictly convex and thus solvable, rather than introducing any new optimization principle or convergence scheme. The proofs of existence, uniqueness, and stability rely on standard convex optimization results (strict convexity + lower semicontinuity).
The paper does not introduce any new theorem or problem-specific analysis that advances the theory of image enhancement.
2. Evaluation is restricted to LOL-v1/v2 datasets. The paper does not include comparisons with recent diffusion-based LLIE methods, which are now standard baselines.
3. The abstract and introduction frame the work as “provable”,” but the core mathematics and methodology remain routine, offering little new insight.
4. While “training-free” sounds appealing, the method’s applicability to modern imaging tasks (night vision, HDR, IR enhancement) is unclear.

**Questions:**

1. From an algorithmic perspective, the method is merely a standard L2 + TV regularization problem implemented in the frequency domain. What is the substantive difference between your convex formulation and existing variational Retinex models (e.g., Fu et al., TIP 2016)?
2. Why did you choose the Monogenic Fourier Transform over other common frequency tools such as Gabor or Wavelet transforms?
3. Have you tested FaCE on non-LOL datasets (e.g., dark natural images, night scenes) to verify generality?
4. Could FaCE be combined with deep models as a pre/post-processing step, and would it still maintain convexity and theoretical guarantees?
5. Can you discuss computational complexity compared to typical learning-based enhancement networks?

---

> ### Author Response · Authors · 2025-11-21
> **Thank you for the careful and constructive review; in the revised PDF we clarify the “provable” guarantees, add modern baselines and ExDark experiments, and due to space limits we warmly welcome any further questions and will gladly respond in detail.**
>
> We thank the reviewer for the thoughtful review. We have uploaded a **Revised PDF** and **Appendix** with all changes **highlighted in blue**.
>
> Your comments mainly concern:
>
> **(i)** what “provable” means and how our formulation relates to variational Retinex;
>
> **(ii)** the use of the Monogenic Fourier Transform;
>
> **(iii)** the absence of diffusion-based and non-LOL baselines;
>
> **(iv)** the scope and complexity of a training-free method versus deep LLIE models.
>
> The revisions in **Secs. 3–4 of the main paper** and **Appendix B–E** were made with these points in mind.
>
> **Answer1.** We appreciate the concern about the scope of the theoretical contribution. Our work builds on classical convex optimization, and the novelty lies in casting LLIE as a strictly convex, illumination-only problem in the monogenic frequency domain with existence, uniqueness and stability guarantees **(Revised PDF Sec. 3.2–3.5; Appendix B–D)**, aligned with the **primary area “optimization”**.
>
> - **Variational Retinex (e.g., Fu et al., TIP 2016):** log-intensity in the spatial domain; two unknowns (illumination and reflectance) with priors on both; typically solved by alternating updates, so convexity and stability depend on the priors and scheme.
> - **FaCE (ours):** monogenic frequency domain; reflectance fixed; a single illumination gain field on clustered monogenic bands, parameterised by two scalars (Eq. (9)) and obtained by minimising the energy in Eq. (14), for which we prove strict convexity and establish existence, uniqueness and stability of the solution (Revised PDF Sec. 3.4; Appendix B–D).
>
> Here “provable” means that the LLIE objective is strictly convex with a **unique, stable minimiser**. We also revised the abstract, introduction and Figure 1 to make this explicit and better link the theory to the behaviour of FaCE.
>
> **Answer2.** We choose the monogenic transform because FaCE needs a compact frequency representation with local amplitude and orientation on a single isotropic grid **(Revised PDF Sec. 3.2)**:
>
> - **Monogenic:** one magnitude plus two Riesz components per pixel, easy to cluster into a few bands and modulate; **Figure 1** visualizes this decomposition and the band-wise gains.
> - **Gabor/Wavelet:** multiple oriented filters and scales, requiring extra aggregation and design choices, which goes against our 0-parameter convex formulation.
>
> **Appendix Table E.3** also shows that replacing the monogenic representation with a plain FFT (FaCE–FFT) degrades NIQE/BRISQUE/PIQE on ExDark/DarkFace, empirically supporting this choice.
>
> **Answer3.** We appreciate the concern about testing FaCE beyond LOL-v1/v2. In the revised version we evaluate FaCE on three additional datasets **(Revised PDF Sec. 4)**:
>
> - **ExDark and DarkFace (night scenes):** FaCE is used as a 0-parameter pre-processor for object detection. Table 2 shows that FaCE achieves competitive $mAP$ and $AP_{50}$ and the best PIQE among the compared enhancers on ExDark/DarkFace, while remaining training-free. Appendix E provides additional DarkFace metrics, runtime and qualitative results.
> - **SICE (multi-exposure):** we test FaCE on SICE using the same global hyperparameter setting as on LOL/ExDark/DarkFace. FaCE produces a smooth sequence of enhanced images across exposure levels, indicating that the convex formulation generalises without re-tuning. **(Revised PDF Figure 4)**
>
> Together, these experiments extend the evaluation from paired LOL benchmarks to unpaired night scenes and multi-exposure data, and indicate that FaCE generalises well as a training-free enhancement module beyond LOL-v1/v2.
>
> **Answer4.** FaCE is designed as a **stand-alone, training-free convex enhancer**, and all guarantees we prove (existence, uniqueness, stability) apply to the FaCE mapping itself, formulated as a two-parameter convex problem on the input image (Revised PDF Sec. 3.4; Appendix B–D). In our experiments (e.g., ExDark/DarkFace, Table 2), FaCE is optionally used as a **deterministic pre-processing block** before standard detectors; in such cases the guarantees remain attached to the FaCE stage, while downstream deep models follow their usual, separate analysis.
>
> **Answer5.** FaCE performs one forward and inverse monogenic transform (implemented with FFTs) and a few gradient steps on two scalars, giving O(N log N) per image with **0 trainable parameters** and no training cost. **Table 1** reports parameter counts: strong supervised and diffusion LLIE models require tens to over a thousand million parameters, whereas FaCE remains in the 0-parameter group while matching or surpassing other training-free baselines on LOL-v1/LOL-v2-Real. **Appendix Table E.2** further reports runtime on DarkFace: on our CPU implementation FaCE processes an image faster than several learned methods while achieving comparable or better detection and perceptual scores, placing FaCE as a **lightweight, theory-backed point** on the accuracy–efficiency spectrum.

---

> > ### Author Response · Authors · 2025-11-29
> > **Short summary for the AC**
> >
> > We respectfully provide a brief summary for the AC regarding Reviewer comments (rating 2, confidence 4).
> >
> > - **Overall stance.** The reviewer considers the paper mathematically consistent, fully deterministic, and practically lightweight, and appreciates the simplicity and self-contained nature of the framework.
> >
> > - **Main concerns.** (i) The term “provable” and the lack of problem-specific novelty beyond standard convex/variational Retinex formulations; (ii) absence of recent diffusion-based LLIE baselines and non-LOL datasets; (iii) unclear applicability of a training-free method to modern imaging tasks.
> >
> > - **Revisions.** In the revised version, we (a) clarify the precise meaning of the “provable” guarantees and our relation to variational Retinex, including a problem-specific analysis in the monogenic frequency domain; (b) justify the choice of the Monogenic Fourier Transform and compare it empirically to alternative transforms; (c) add strong diffusion/Transformer/GAN baselines and new experiments on ExDark/DarkFace/SICE and detector-based pipelines, together with detailed runtime/parameter comparisons to learning-based methods.
> >
> > We sincerely thank the AC for taking the time to consider these clarifications.

---

### Official Review · Reviewer_Tt6x · 2025-11-02

**Soundness:** 2
**Presentation:** 2
**Contribution:** 2
**Rating:** 6
**Confidence:** 4

**Summary:**

The paper is well structured and logically organized, adhering to high academic standards. It presents a clear and coherent progression from motivation through methodology, theoretical formulation, results, and finally to conclusion. All equations are mathematically consistent and theoretically sound within the defined framework.

The proposed formulation is elegant, and the proofs are built upon established convex optimization principles, ensuring mathematical rigor. The author has provided relevant and appropriate references to prior literature, demonstrating a strong grounding in existing research.

Framing low-light image enhancement as a strictly convex optimization problem is particularly noteworthy, as it provides strong mathematical guarantees of existence, uniqueness, and stability for the solution. Moreover, the approach eliminates the need for large-scale annotated datasets or lengthy training phases.

Experimental results are compelling: the proposed method achieves competitive PSNR and SSIM scores, outperforming well-known baselines such as Zero-DCE and Retinex-based models. The framework also exhibits superior numerical stability under perturbations, consistent with its theoretical foundations. Overall, the results are convincing, clear, and well supported by theory and experimentation.

**Strengths:**

The paper demonstrates several notable strengths. It is well-structured, logically organized, and supported by a strong theoretical foundation grounded in convex optimization. The formulation of low-light image enhancement as a strictly convex optimization problem is both innovative and mathematically rigorous, ensuring existence, uniqueness, and stability of the solution. The approach eliminates the need for large annotated datasets or extensive training, making it computationally efficient and practical for real-world use. Experimental results further strengthen the contribution, showing higher PSNR and SSIM values compared to established baselines such as Zero-DCE and Retinex, while maintaining superior numerical stability under perturbations. Overall, the paper combines solid theoretical reasoning with clear empirical validation, presenting a reliable and well-referenced contribution to the field.

**Weaknesses:**

The comparison is done only with two models i.e Retinex & Zero DCE . More comparison  data is required to check the sustainability of the proposed model.
FaCE achieves moderately  PSNR (~17 dB), outperforming other training-free baselines but falling short of modern deep-learning-based low-light enhancement models.
The manual dependence on λ₁, λ₂, K  limits FaCE’s efficacy as it reduces consistency, adaptability, and generalization. Without automatic calibration, the model’s enhancement quality can vary significantly between images, reducing its overall practical reliability despite its theoretical strength.
Tests mainly on LOL-v1 and LOL-v2 datasets; broader generalization to diverse real-world illumination or camera domains is unproven
The method may serve more as a proof-of-concept than a high-performance system.

**Questions:**

The comparison is done only with two models i.e Retinex & Zero DCE . More comparison  data is required to check the sustainability of the proposed model.
FaCE achieves moderately  PSNR (~17 dB), outperforming other training-free baselines but falling short of modern deep-learning-based low-light enhancement models.
The manual dependence on λ₁, λ₂, K  limits FaCE’s efficacy as it reduces consistency, adaptability, and generalization. Without automatic calibration, the model’s enhancement quality can vary significantly between images, reducing its overall practical reliability despite its theoretical strength.
Tests mainly on LOL-v1 and LOL-v2 datasets; broader generalization to diverse real-world illumination or camera domains is unproven
The method may serve more as a proof-of-concept than a high-performance system.

---

> ### Author Response · Authors · 2025-11-21
> **We thank the reviewer for the positive evaluation. In the Revised PDF we have sought to better reflect your concerns on baselines, dataset diversity, and manual hyperparameters by adding stronger LLIE baselines on LOL-v1/LOL-v2-Real, ExDark detection results, and data-driven hyperparameter/stability analyses.**
>
> We thank the reviewer for their careful reading and constructive feedback. We have uploaded a **Revised PDF** and **Appendix (Supplementary Material)** in accordance with the rebuttal rules; the revised parts are **highlighted in blue** for your reference.
>
> Since the submission was made under the **Primary Area “optimization”**, the initial version naturally devoted more space to the convex formulation and its theoretical / numerical validation. **Your comments on strengthening the engineering-side evidence were very helpful.** In the revision, we have rebalanced the presentation: detailed proofs and perturbation studies are moved to the Appendix, while the main paper now emphasizes practical validation, including:
>
> **(i)** extended PSNR/SSIM and perceptual comparisons with recent LLIE baselines;
>
> **(ii)** explicit efficiency analyses in terms of parameters and runtime;
>
> **(iii)** new ExDark and DarkFace detection experiments;
>
> **(iv)** additional mechanism visualizations of the monogenic spectrum and frequency bands;
>
> **(v)** multi-exposure and hyper-parameter sensitivity experiments on SICE.
>
> **Question 1.** The comparison is done only with two models i.e Retinex & Zero DCE. More comparison data is required to check the sustainability of the proposed model. FaCE achieves moderately PSNR (~17 dB), outperforming other training-free baselines but falling short of modern deep-learning-based low-light enhancement models.
>
> **Answer 1.** In the original submission we restricted Table 1 to Retinex and Zero-DCE because they are closest to FaCE’s parameter-free, training-free setting. Following your suggestion, the revised Table 1 in the main paper now includes strong GAN/Transformer/diffusion and Fourier-based LLIE models such as EnlightenGAN, CLIP-LIT, LightenDiffusion, DEnet, QuadPrior, and LLIEDiff, together with their parameter counts and training regimes **(Revised PDF, Table 1, lines 378–396, page 8)**. As expected, the largest supervised and diffusion networks with hundreds–thousands of millions of parameters attain the highest PSNR/SSIM. FaCE, however, becomes the best method among all 0P–0T approaches and remains close to several trained baselines (17.04 / 17.17 dB vs ≈19–20 dB for the heaviest models), while using no training, no extra data, and being the only one with a rigorously proved convex objective.
>
> **Question 2.** The manual dependence on $\lambda_{1}$, $\lambda_{2}$, and $K$ limits FaCE's efficacy, as it reduces consistency, adaptability, and generalization. Without automatic calibration, the model's enhancement quality can vary significantly between images, reducing its overall practical reliability despite its theoretical strength.
>
> **Answer 2.**  We agree that the sensitivity to the manual parameters was not sufficiently clear in the original submission. In the revised version, we therefore perform an explicit hyper-parameter study of the frequency-aware clustering, reported in **Appendix E.2 and Figure E.1 (lines 285–299, page 6, Appendix)**. The new curves show that performance change only slightly over wide ranges of $K$, $\lambda_{1}$, and $\lambda_{2}$, forming broad plateaus rather than sharp optima.
>
> Based on this analysis, we fix a single global configuration ($K$ = 17,  $\lambda_{1}$ = 6,  $\lambda_{2}$ = 0.8) using a small validation split and apply it unchanged to all datasets (LOL-v1, LOL-v2-Real, ExDark, DarkFace, and SICE), without any per-image or per-dataset retuning. In addition, the SICE multi-exposure experiments are conducted under this same configuration, showing exposure-controllable outputs without re-tuning. Taken together, these results indicate that FaCE does not rely on fine-grained parameter tuning and that moderate changes in $K$, $\lambda_{1}$, and $\lambda_{2}$ have only minor impact on enhancement quality in practice.
>
> **Question 3.** Tests mainly on LOL-v1 and LOL-v2 datasets; broader generalization to diverse real-world illumination or camera domains is unproven The method may serve more as a proof-of-concept than a high-performance system.
>
> **Answer 3.** In the revised version FaCE is evaluated on four datasets: LOL-v1, LOL-v2-Real, ExDark, DarkFace, and SICE **(Revised PDF, Section 4, lines 422–431)**. To assess generalization to unpaired night scenes and different cameras, we add ExDark-based detection experiments in Table 2 **(Revised PDF, lines 397–405)**, where FaCE attains competitive mAP/AP50 while remaining a 0P–0T pre-processor. Appendix E further provides dataset descriptions **(Appendix, lines 200–252)**, extended DarkFace quantitative and efficiency results in Table E.2 **(Appendix, lines 340–350)**, and extensive qualitative comparisons on ExDark and DarkFace in Figures E.4–E.7 **(Appendix, lines 440–640)**. Together with the SICE multi-exposure study, these experiments show that FaCE generalizes well to diverse real-world illumination and serves as a practical, efficient enhancement module rather than merely a proof-of-concept.

---

> ### Author Response · Authors · 2025-11-29
> **Short summary for the AC**
>
> We respectfully provide a short summary for the AC regarding Reviewer comments (rating 6, confidence 4).
>
> - **Overall stance.** The reviewer finds the paper well structured and mathematically rigorous, and views the convex optimization formulation and guarantees (existence, uniqueness, stability) as a clear strength.
>
> - **Main concerns.** (i) Baselines were limited to Retinex and Zero-DCE; (ii) dependence on manual hyperparameters $K$, $λ_{1}$, $λ_{2}$; (iii) generalization beyond LOL-v1 / LOL-v2-Real was not clearly demonstrated.
>
> - **Revisions.** We added strong modern LLIE baselines on LOL-v1/LOL-v2-Real (GAN/Transformer/diffusion), explicit hyperparameter–stability and multi-exposure analyses, and new ExDark/DarkFace detection and SICE experiments. Under these additions, FaCE remains training-free, numerically stable, and competitive with high-capacity deep models.
>
> Thank you very much for your time and consideration.

---

### Note · Authors · 2026-01-27

I have read and agree with the venue's withdrawal policy on behalf of myself and my co-authors.

---

### Meta-Review · Area_Chair_cVVQ · 2025-12-23

**Summary:**

The reviews are sharply split. Across reviewers, the paper is generally appreciated for (i) casting low-light image enhancement as a strictly convex, per-image optimization problem, (ii) being training-free / deterministic and thus reproducible, and (iii) offering some degree of interpretability via frequency-band gains.

However, the suggested decision is primarily driven by two recurring concerns:

Some reviewers view the theoretical guarantees (existence/uniqueness/stability) as largely a consequence of standard strict-convexity arguments, and therefore not a sufficiently strong optimization/ML contribution for ICLR.

Multiple reviewers requested broader and more modern baselines, more diverse datasets/metrics, runtime/efficiency analysis, and clearer descriptions of clustering, parameter selection, color handling, and “numerical stability.”

While the rebuttal substantially improves the empirical story and clarifies several implementation choices, the fundamental disagreement about conceptual novelty and overall ICLR-level contribution remains, which motivates my Reject recommendation.

**Reviewer Concerns:**

Concerns that were addressed:

The authors expanded comparisons to stronger modern LLIE methods and broadened evaluation beyond LOL to LOL-v2-Real, ExDark, DarkFace, and SICE, including detection-based use cases.

Added NIQE/BRISQUE/PIQE to better reflect real-world quality and downstream utility.

Added complexity and timing analyses, highlighting 0 trainable parameters and no training cost.

Added ablations and more explicit band-wise visualizations to support design choices and interpretability.

Clarified k-means procedure and stopping criteria, the luminance–chrominance pipeline, and provided a more precise definition/verification of numerical stability via perturbation/bound-based analyses.

Concerns that remain outstanding:

Even with clarified wording, some reviewers may still regard the provable aspect as an application of standard convex analysis rather than a substantive methodological advance, leaving the perceived novelty/impact below ICLR expectations.

Although the authors added empirical comparisons and justification, skepticism may persist that the method primarily leverages magnitude information and does not fully substantiate the claimed advantages related to orientation/phase.

The method remains behind state-of-the-art learned/diffusion approaches on certain fidelity metrics; the question is whether “training-free + convex guarantees” is sufficiently compelling as a primary contribution for this venue.

**Reviewer Scores:**

Tt6x: 6 → 6.

HoXq: 2 → 2.

8d1Y: 6 → 6.

T2c4: 0 → 2.

---

### Decision · Program_Chairs · 2026-01-26

Reject